# Mammalian cells measure the extracellular matrix area and respond through switching the adhesion state

Xiaole Wang [1], Pengli Wang [1], Lihang Zhang[1], Tianyu Xu [1], Seungkuk Ahn [1,2], Upnishad Sharma[1], Han Yu [1], Nico Strohmeyer [1] ✉ & Daniel J. Müller [1] ✉

Mammalian cells adjust integrin-mediated adhesion based on the composition and structure of the extracellular matrix (ECM). However, how spatially confined ECM ligands regulate cell adhesion initiation remains unclear. Here, we investigate how cells adapt early adhesion to different ECM protein areas. Through combining microcontact printing with single-cell force spectroscopy we measure cell adhesion initiation and strengthening to defined areas of ECM proteins. HeLa cells and mouse embryonic fibroblasts gradually increase adhesion with collagen I or fibronectin area, while reaching maximum adhesion force to ECM patterns having areas above certain thresholds. On much smaller patterns, both cell types switch to a different state and considerably increase the adhesion force per ECM protein area, which they strengthen much faster. This spatially enhanced adhesion state does not require talin or kindlin, indicating a fundamentally different adhesion mechanism. Mechanotransduction seems to play integrin and cell type-specific roles in the spatially enhanced adhesion state.

Cell adhesion to the extracellular matrix (ECM) plays an integral role in cell morphology and function and is of fundamental importance in tissue morphogenesis and maintenance[1–3]. While all ECMs are composed of diverse proteins and polysaccharides, each tissue possesses specific ECM properties evolving from dynamic interactions between the various cells within the tissue and the proteinaceous microenvironment[4,5]. Cells employ α/β heterodimeric integrins to adhere to ECM proteins and to respond to the biochemical and biophysical properties of the ECM[6–8]. In mammals, the integrin superfamily encompasses 24 members, many of which being co-expressed on the cell surface[9]. The dynamic binding and unbinding of integrins to and from their ECM ligands are tightly regulated through conformational changes[10] and essential for a variety of cellular processes, such as migration[11,12] and cell division[13,14]. The cytoplasmic adaptor proteins kindlin and talin, which bind to the cytoplasmic domain of the integrin β-subunit, are essential for integrin-mediated cell adhesion[15–19]. Kindlin diffuses along the cell membrane[20], binds to the cytoplasmic tail of the integrin β-subunit, increases the binding affinity of talin to the β-subunit[21], and recruits various other proteins, including paxillin[19,22]. Talin consists of an integrin-activating talin head (FERM) domain that binds the β-subunit, and the talin rod domain that engages the integrin to the actin cytoskeleton to allow actomyosin mediated force transduction[23,24]. After ligand binding, integrins cluster to form nascent adhesions that either dismantle or mature into large and long-lived focal adhesions through talin-transduced actomyosin-mediated mechanical force[25]. Thereby, paxillin supports integrin clustering and adhesion maturation by recruiting more adhesome proteins to the adhesion site[26,27].

Cells in tissues are susceptible to their structured environment, which originates from the location, orientation, and dimensionality of the ECM[28]. This structural information dictates the spatial distribution,

[1]Eidgenössische Technische Hochschule (ETH) Zurich, Department of Biosystems Science and Engineering, Klingelbergstrasse 48, Basel, Switzerland.
[2]Present address: Charles Institute of Dermatology, School of Medicine, University College Dublin, Dublin, Ireland. ✉e-mail: nico.strohmeyer@bsse.ethz.ch; daniel.mueller@bsse.ethz.ch

size, and maturation state of the adhesion sites[29]. In vitro integrin ligand-nanopatterning demonstrated that the clustering of integrins and focal adhesion maturation depend on the spatial distribution of integrin ligands. This maturation of focal adhesions requires integrin-ligands to be closer than 60 nm (Ref. 30–34). If integrin ligands are patterned in parallel lines, they must be either at least 40 nm thick or the interline distances must be below 200 nm for narrower lines of ligands[35]. More complex organizations with crossing lines of ECM ligands that mimic more complex architectures of the ECM allow focal adhesion maturation on lines of ligands with only 10 nm width. Furthermore, micropatterning approaches have shown that the geometry of ECM protein coated areas can determine the localization of adhesion sites and the morphology of the cytoskeleton, cell contractility, and the segregation of integrins within focal adhesions[36–39]. Yet, how cells that initiate adhesion sense and respond to spatially confined ECMs, such as provided by confined areas of ECM ligands, remains elusive.

Single-cell force spectroscopy (SCFS) describes a family of powerful nanotools that allow to quantify the initiation of cell adhesion at high time and force resolution[40]. Usually, SCFS brings single cells into contact with a substrate of interest, which can be an inorganic surface, biomaterial, another cell, or tissue, to establish adhesive interactions for a defined contact time. Upon mechanically separating the cell from the substrate, the adhesion force and energy of the whole cell, as well as the (un-)binding of single cell adhesion receptors, can be quantified. Among all SCFS tools, atomic force microscopy (AFM)-based SCFS offers the widest force range from ≈20 pN to ≈100 nN (Refs. 41–43). AFM-based SCFS is widely used to quantify the ligand-binding of individual integrins[44–46], the initiation of cell adhesion to different ECM proteins[47], the mechanotransduction during initiating cell adhesion[8,48], the cell adhesion regulation throughout the cell cycle[49,50], the crosstalk of cell adhesion receptors[51–53], the cell adhesion regulation to micro- and nanostructured surfaces[54,55], or the involvement of adhesome proteins in regulating cell adhesion initiation[19,56]. However, the adhesion of cells strongly depends on their contact area with the ECM, which scales with the availability of cell adhesion receptors that can bind ECM ligands and collectively contribute to cell adhesion. So far, no approach has been introduced to confine the contact area of cells initiating adhesion and to quantify the cell adhesion force established. Such correlative measurement would allow to characterize whether and how cells measure and regulate adhesion in response to the ECM area. To address this challenge, here we produce ECM patterns of defined shapes and areas using microcontact printing (µCP) and

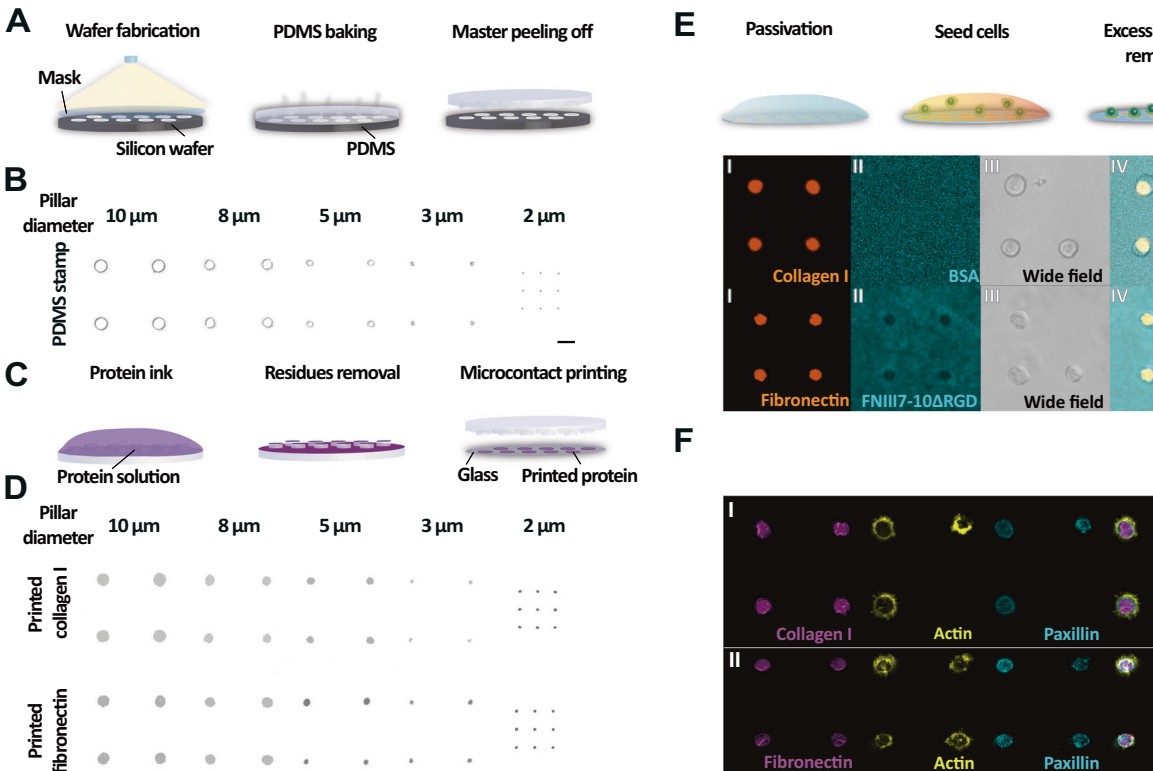

**Fig. 1 | Engineering ECM protein patterns with defined sizes by microcontact printing (µCP). A** Illustration of the production of polydimethylsiloxan (PDMS) stamps having micropillars of different diameters. First, a silicon wafer with arrays of circular holes with 10 µm, 8 µm, 5 µm, 3 µm, or 2 µm diameter is fabricated by photolithography. Then, PDMS is poured onto the wafer, cured, and subsequently the PDMS stamp (master) with micropillars is peeled off from the wafer. **B** Differential interference contrast (DIC) microscopy of the top layer of PDMS stamps with arrays of micropillars of different diameters. Scale bar, 20 µm. **C** µCP-based patterning of ECM proteins on glass surfaces. First, the PDMS stamp is coated (inked) with fluorescently labeled (Alexa fluor 555) ECM proteins (left), afterwards washed with PBS and deionized water to remove residues, then blow-dried by air (middle), and finally ECM proteins are printed on glass through physical contact (right). **D** Confocal microscopy images of circular collagen I (top) and fibronectin (bottom) patterns printed with PDMS stamps having arrays of micropillars of different diameters. Scale bars, 20 µm. **E** Fluorescence images of (I) Alexa fluor 555-labeled collagen I (top) or fibronectin (bottom) patterns printed on glass using PDMS micropillars having 10 µm diameter. (II) After printing of ECM proteins, the uncoated glass surfaces are passivated with FITC-labeled BSA (top) or Alexa fluor 488-labeled FNIII7-10ΔRGD (bottom). (III) Then, wild type (wt) HeLa cells (top) or wt fibroblasts (bottom) are seeded onto printed collagen I or fibronectin patterns for 60 min. Afterwards, weakly attached HeLa cells or fibroblasts are removed by gentle washing with PBS. (IV) merge of (I), (II) and (III). Scale bars, 20 µm. **F** Representative confocal microscopy images of fixed (I) wt HeLa cells adhering to collagen I patterns (pink) or (II) wt fibroblasts adhering to fibronectin patterns (pink) labeled for actin (yellow) and paxillin (blue). Scale bars, 15 µm. n = 10 independent experiments for each condition.

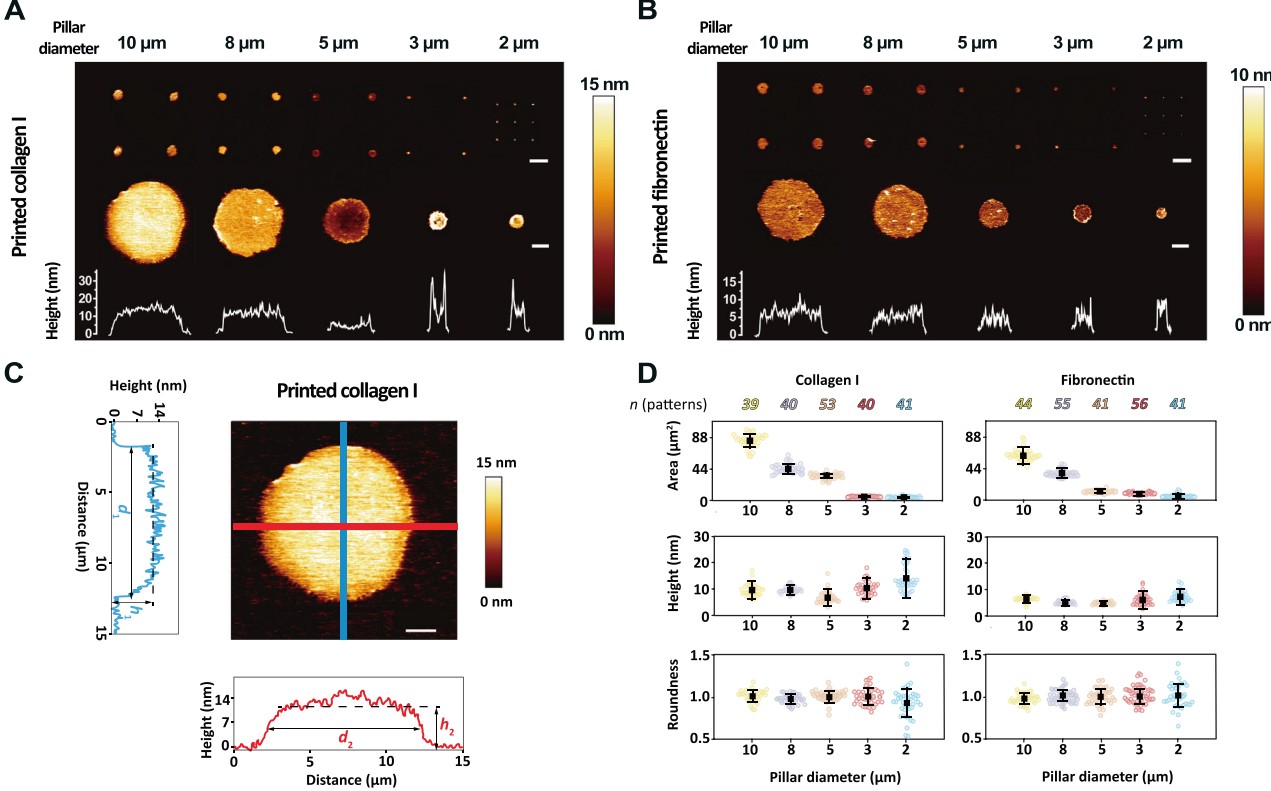

**Fig. 2 | Morphological characterization of ECM protein patterns.** AFM topographs (height images) of (**A**) collagen I or (**B**) fibronectin patterns printed on glass with PDMS stamps having arrays of micropillars of different diameters. Top row shows topographs of 2 ×2 or 3 ×3 arrays of printed ECM protein patterns. Scale bar, 20 μm. Middle and bottom rows show the printed patterns at higher magnification and the height profile of the cross section of (**A**) collagen I or (**B**) fibronectin patterns. Scale bar, 2 μm. Images were acquired form at least 5 independent preparations. **C** Perpendicular diameters ($d_1$, $d_2$) and heights ($h_1$, $h_2$) of a printed ECM protein pattern ($n = 39$, 40, 53, 40, or 41 of printed collagen I patterns having diameters of 10 μm, 8 μm, 5 μm, 3 μm, or 2 μm; $n = 44$, 55, 41, 56, or 41 of printed fibronectin patterns having diameters of 10 μm, 8 μm, 5 μm, 3 μm, or 2 μm, respectively), used to quantify the average area, height, and roundness of the printed pattern. Scale bar, 2 μm. **D** Average area (top), height (middle), and roundness (bottom) derived from $n$ number of collagen I or fibronectin patterns, which have been printed using PDMS micropillar of different diameters. Colored dots represent values of individual printed patterns, black dots mean values, and error bars standard deviations. Statistical analysis of the patterns is given in Supplementary Tables 1, 2.

quantify how mammalian cells initiate and regulate adhesion to ECM patterns using AFM-based SCFS.

## Results

### Engineering patterned ECM substrates of defined area

To create defined areas of ECM proteins, we used μCP to print arrays of circular collagen I patterns to which HeLa (Kyoto) cells can adhere via α1β1 and α2β1 integrins[57] and circular fibronectin patterns to which fibroblasts can adhere via α5β1 and αV-class integrins[39]. For this purpose, we engineered stamps of poly(dimethyl)siloxane (PDMS) micropillars from silicon molds having circular holes with diameters of 10 μm, 8 μm, 5 μm, 3 μm, or 2 μm (Fig. 1A, B). Next, we inked the PDMS micropillars with fluorescently labeled collagen I or fibronectin (Fig. 1C) and printed them in circular patterns onto glass surfaces through physical contact (Fig. 1D). To prevent cells from unspecific attachment, we passivated the non-patterned glass surface using fluorescently labeled bovine serum albumin (BSA) for printed collagen I or a fibronectin fragment lacking the integrin-binding RGD domain (FNIII7-10ΔRGD)[52] for printed fibronectin (Fig. 1E). We then seeded wild type (wt) HeLa cells or mouse embryonic kidney fibroblasts onto collagen I or fibronectin patterns, which we printed using PDMS micropillars of 10 μm diameter. We allowed the cells to adhere for 60 min and removed non-attached cells by gentle washing. Confocal microscopy confirmed that HeLa cells and fibroblasts only adhered to printed collagen I or fibronectin patterns, respectively. We further stained HeLa cells and fibroblasts for actin and paxillin, a marker for integrin-mediated adhesions (Fig. 1F)[58,59]. HeLa cells

and fibroblasts did not form stress fibers or focal adhesions, indicting that cells seeded on small ECM protein patterns cannot break symmetry as they do on much larger circular ECM patterns[39,60].

### Characterization of ECM protein patterns

Next, we characterized the circular ECM protein patterns printed on glass by contact mode AFM (Fig. 2A, B). AFM topographs showed protein patterns protruding from the glass. On average, collagen I patterns printed with micropillars having diameters of 10 μm, 8 μm, 5 μm, 3 μm, or 2 μm showed heights of $9.7 \pm 2.3$ nm (mean ± SD), $9.8 \pm 1.2$ nm, $6.2 \pm 2.1$ nm, $10.4 \pm 2.6$ nm, or $14.2 \pm 5.0$ nm ($n = 39$, 40, 53, 40, 41; Supplementary Table 1), while printed fibronectin patterns showed heights of $6.5 \pm 1.0$ nm, $5.1 \pm 0.8$ nm, $4.9 \pm 0.6$ nm, $6.0 \pm 2.0$ nm, or $7.3 \pm 2.0$ nm ($n = 44$, 55, 41, 56, 41; Supplementary Table 2). Due to the mechanical printing process, the ECM patterns were not perfectly circular and showed irregular edges. The area covered by ECM proteins printed with micropillars having diameters of 10 μm, 8 μm, 5 μm, 3 μm, or 2 μm was $81.2 \pm 9.2$ μm², $42.6 \pm 6.2$ μm², $33.2 \pm 3.6$ μm², $4.3 \pm 0.8$ μm², or $3.4 \pm 1.1$ μm² for collagen I patterns and $58.1 \pm 7.7$ μm², $34.1 \pm 4.6$ μm², $8.8 \pm 1.9$ μm², $5.3 \pm 1.4$ μm², or $2.4 \pm 2.2$ μm² for fibronectin patterns (Fig. 2C, D; Supplementary Fig. 1A, B).

### Cells initiating adhesion respond to the area of printed ECM proteins

The area of the printed ECM protein pattern directly impacts the contact area between the cell and the ECM substrate and thus limits

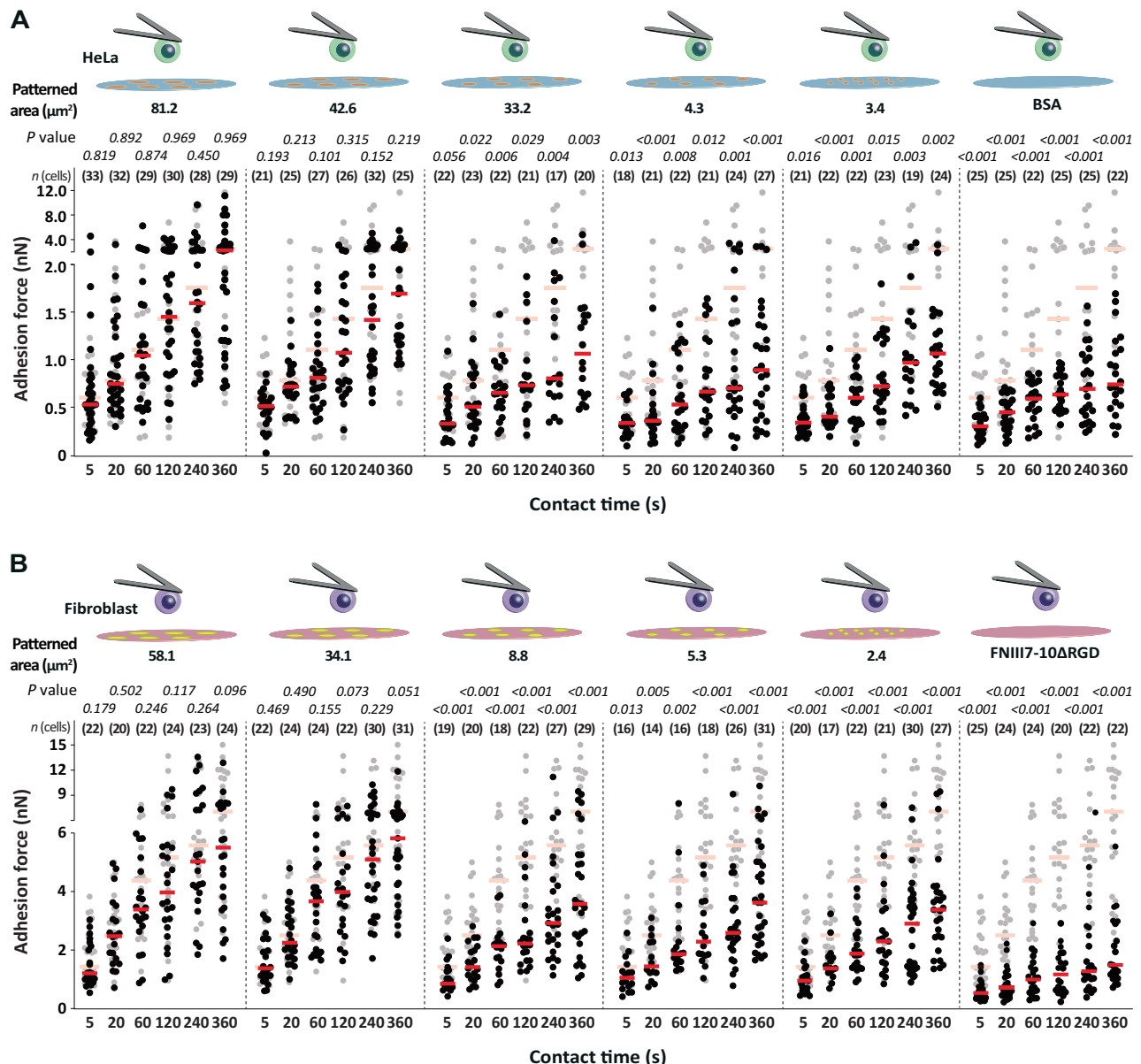

**Fig. 3 | Adhesion force of HeLa cells and fibroblasts correlates with the area of the patterned ECM protein substrate.** Adhesion force of (**A**) wt HeLa cells and (**B**) wt fibroblasts to collagen I and fibronectin patterns having different areas and measured at different contact times. After printing the ECM protein patterns, the remaining uncoated glass surfaces were passivated with (**A**) BSA or (**B**) FNIII7-10ΔRGD to minimize unspecific cell adhesion to glass. Dots represent the adhesion force of single cells, red bars median values, and n (cells) the number of independent cells tested in at least three independent experiments. Adhesion force of (**A**) wt HeLa cells or (**B**) wt fibroblasts to unrestricted collagen I or fibronectin is given in semitransparent as reference. P values were calculated using two sided Mann–Whitney tests and compare cell adhesion forces of displayed data (black) and reference data. Statistical analysis of potential difference in adhesion forces established by wt Hela cells to collagen I patterns and wt fibroblasts to fibronectin patterns of different sizes is given in Supplementary Tables 3, 4.

the number of integrin-binding sites, which are available to initiate adhesion[33]. To characterize the impact of the ECM protein pattern area on cell adhesion initiation, we characterized the adhesion force of wt HeLa cells and fibroblasts by AFM-based SCFS (Supplementary Fig. 1C). Thereto, we attached a single rounded wt HeLa cell or wt fibroblast to concanavalin A (ConA)-coated microcantilevers[61] and positioned it above an individual fluorescently labeled ECM protein pattern (Supplementary Fig. 1D). We then allowed the cell to initiate and strengthen adhesion to individual collagen I or fibronectin patterns for contact times ranging from 5 to 360 s before quantifying the cell adhesion force (Fig. 3). Importantly, by combining SCFS and confocal microscopy we showed that the contact area between the cell and the substrate was independent of the substrate, always larger than the ECM protein pattern, and only marginally increased over contact time,

which did not affect SCFS experiments (Supplementary Fig. 1E, F). Hence, the area to which cells could initiate integrin-mediated adhesion was restricted by the printed ECM pattern. As a control, we measured the adhesion force of HeLa cells to unrestricted collagen I or BSA substrates and of fibroblasts to unrestricted fibronectin or FNIII7-10ΔRGD substrates.

Wt HeLa cells and wt fibroblasts established minimal adhesion force to BSA and FNIII7-10ΔRGD, respectively (Fig. 3), thus confirming that the passivation of the non-patterned glass surface with either BSA or FNIII7-10ΔRGD suppressed unspecific cell adhesion to glass[50,62]. The adhesion force of wt HeLa cells to all printed collagen I patterns or of wt fibroblasts to all printed fibronectin patterns was higher compared to BSA or FNIII7-10ΔRGD, verifying that both cell lines established integrin-mediated adhesion. Furthermore, both cell lines increased the

adhesion force to collagen I or fibronectin patterns with contact time. HeLa cells established similar adhesion force to unrestricted collagen I substrates and printed collagen I patterns having areas of ≈ 81.2 μm² and ≈ 42.6 μm², which verified that the collagen I printing process did not affect adhesion initiation (Fig. 3A; Supplementary Fig. 2A). However, HeLa cells considerably reduced the adhesion force to collagen I patterns having areas ≤33.2 μm². HeLa cells established similarly low adhesion force to collagen I patterns having areas of ≈ 33.2 μm², ≈ 4.3 μm² and ≈ 3.4 μm² (Supplementary Table 3). Fibroblasts established similar adhesion force to unrestricted fibronectin substrates and fibronectin patterns having areas of ≈ 58.1 μm² and 34.1 μm² (Fig. 3B; Supplementary Fig. 2B). However, they considerably reduced adhesion force to smaller fibronectin patterns of ≈ 8.8 μm², ≈ 5.3 μm² or ≈ 2.4 μm², which were similarly low (Supplementary Table 4).

The results demonstrate that cells differentially initiate adhesion in response to the area of ECM protein and, hence, to the availability of integrin binding sites provided by the ECM protein pattern. When the contact area to collagen I is ≥ 42.6 μm² HeLa cells establish maximum adhesion force to collagen I. However, if the contact area to collagen I is smaller, the cells establish lower adhesion force. Similarly, fibroblasts in contact to fibronectin patterns ≥ 34.1 μm² establish maximum adhesion force, while they establish lower adhesion force if the area of the fibronectin pattern is smaller.

## With reducing ECM protein area cells switch to a spatially enhanced adhesion state

Next, we investigated whether the adhesion force per ECM protein pattern area of wt HeLa cells and wt fibroblasts correlates with the ECM protein pattern size (Fig. 4A, B). Fibroblasts established higher adhesion force per fibronectin area than HeLa cells per collagen I area, which may be related to the different cell type, ECM substrate, and/or different integrins binding the different substrates. However, the adhesion force per area inversely correlated with the overall area of the printed ECM protein pattern. Whereas HeLa cells established similar adhesion force per area to collagen I patterns having areas of ≈ 81.2 μm², ≈ 42.6 μm² and ≈ 33.2 μm², they increased the adhesion force per collagen I area 6- to 8-fold to ≈ 4.3 μm² large collagen I patterns and 9- to 15-fold to ≈ 3.4 μm² large collagen I patterns (Fig. 4A). Fibroblasts established similar adhesion force per area to ≈ 58.1 μm² and ≈34.1 μm² large fibronectin patterns and increased the adhesion force per area to ≈ 4-fold to ≈ 8.8 μm², 5- to 7-fold to ≈ 5.3 μm², and 10- to 15-fold to ≈ 2.4 μm² large fibronectin patterns (Fig. 4B). Thereby, the increase of cell adhesion force per area to smaller compared to larger ECM protein patterns remained similar over contact time (Supplementary Fig. 3).

To consider possible unspecific contributions to the cell adhesion, we subtracted the unspecific adhesion force to either BSA or FNIII-10ΔRGD to correct for the specific cell adhesion force per ECM protein area (Supplementary Fig. 4). This correction did not impact the observation that wt HeLa cells establish low corrected adhesion force per area for collagen I patterns ≥33.2 μm² and drastically higher adhesion force per area to the smaller collagen I patterns. Further, also wt fibroblasts established low corrected adhesion force per fibronectin area to patterns ≥34.1 μm², and drastically higher adhesion forces per area to the smaller fibronectin patterns (Supplementary Fig. 4). The control, thus, confirmed that the unique adhesion force per area increase with decreasing ECM protein area is related to specific cell adhesion established to ECM proteins.

Next, we quantified the unbinding force of single integrins from the ECM ligand (Supplementary Fig. 5A). First, we evaluated the unbinding force of single integrins of wt HeLa cells adhering to unrestricted collagen I and of wt fibroblasts adhering to unrestricted fibronectin across all contact times (Supplementary Fig. 5B, C). The unbinding force derived from Gaussian fits revealed 37.15 ± 10.15 pN (mean ± SD) for HeLa cells and 44.73 ± 10.40 pN for fibroblasts. In comparison, the unbinding force of single unspecific binding events of HeLa cells from

BSA (31.87 ± 7.53 pN) and of fibroblasts from FNIII7-10ΔRGD (36.71 ± 9.73 pN) were considerably lower. The specific force recorded of single integrins unbinding from ECM ligands and the unspecific unbinding force agree well with previous reports[43,63–65]. We further tested the single integrin unbinding force of HeLa cells or fibroblasts adhering to differently sized collagen I and fibronectin patterns, which all showed similar values as those measured using unrestricted collagen I or fibronectin substrates. This finding indicates that the area of the ECM protein pattern does not change the ligand-binding strength of single integrins.

The quantification of the adhesion force per patterned ECM protein area, which was corrected for contribution of the unspecific adhesion, and the unbinding force of single integrins allowed us to estimate the number of integrins binding to ECM ligands (Supplementary Fig. 6). For this, we divided the corrected adhesion force per area by the mean rupture force of single integrins. However, due to the spring-like loading of cantilever and cell during the mechanical detachment process (Supplementary Fig. 5A)[40–42,66,67], our estimation likely underestimates the number of rupture events and thus of ligand-bound integrins. We deliberately considered tether events for estimating the number of ligand-bound integrins since single membrane tethers are linked to the ECM protein by at least one integrin[43,67,68]. However, since the forces of tether events are higher than those of rupture events, our approach likely overestimates the number of ligand-bound integrins in the part of the force-distance curve that is dominated by tether events (under the assumption that one integrin links the tether to the substrate). Despite these limitations, the estimated density of ligand-bound integrins increased on all collagen I patterns with contact time, verifying that the estimation provides reasonable results (Supplementary Fig. 6A). The estimated number of integrin-ligand bonds wt HeLa cells established per ECM protein area was very low (<0.7 μm⁻²) to collagen I patterns of ≈ 81.2 μm², ≈ 42.6 μm² and ≈ 33.2 μm² for all contact times. However, this number increased to ≈1–6 μm⁻² to collagen I patterns having smaller areas of ≈ 4.3 μm² and ≈ 3.4 μm². Similarly, the estimated number of integrin-ligand bonds fibroblasts established per area increased gradually with decreasing area of the printed fibronectin pattern (Supplementary Fig. 6B). Whereas on ≈ 58.1 μm² and ≈ 34.1 μm² large fibronectin patterns the number of ligand-integrin bonds per area was <4 μm⁻² for all contact times, they increased to ≈ 11 μm⁻² on fibronectin patterns of ≈ 5.3 μm² and ≈ 2.4 μm² at 360 s contact time.

In summary, when the area of ECM proteins falls below a certain threshold, HeLa cells and fibroblasts transition from their canonical adhesion state to a spatially enhanced adhesion state, which is characterized by both a considerably increased adhesion force per ECM protein area and a considerably increased estimated density of ligand-bound integrins.

## The spatially enhanced adhesion state accelerates adhesion strengthening

Next, we wanted to understand how adhesion strengthening, that is, the adhesion force increase with contact time, of wt HeLa cells and wt fibroblasts depends on the area of the patterned ECM protein. Hence, we quantified the adhesion strengthening rate as the slope of a linear fit through the adhesion force or the adhesion force per area for all contact times (Fig. 4C). As expected, HeLa cells and fibroblasts strengthen adhesion minimally to BSA or FNIII7-10ΔRGD. HeLa cells strengthened adhesion similarly to unrestricted collagen I and collagen I patterns having areas of ≈ 81.2 μm² and ≈ 42.6 μm². Similarly, fibroblasts strengthened adhesion to unrestricted fibronectin substrates and to patterned fibronectin substrates having areas of ≈ 58.1 μm² and 34.1 μm² at similar rates. HeLa cells and fibroblasts strengthened adhesion at lower rates to ECM protein patterns having smaller areas. However, for both cell lines the adhesion strengthening per area inversely correlated to the ECM protein area (Fig. 4D). Although HeLa cells strengthened adhesion per area to ≈81.2 μm², ≈42.6 μm², and ≈33.2 μm² collagen I patterns at similarly low rates, they drastically increased their adhesion

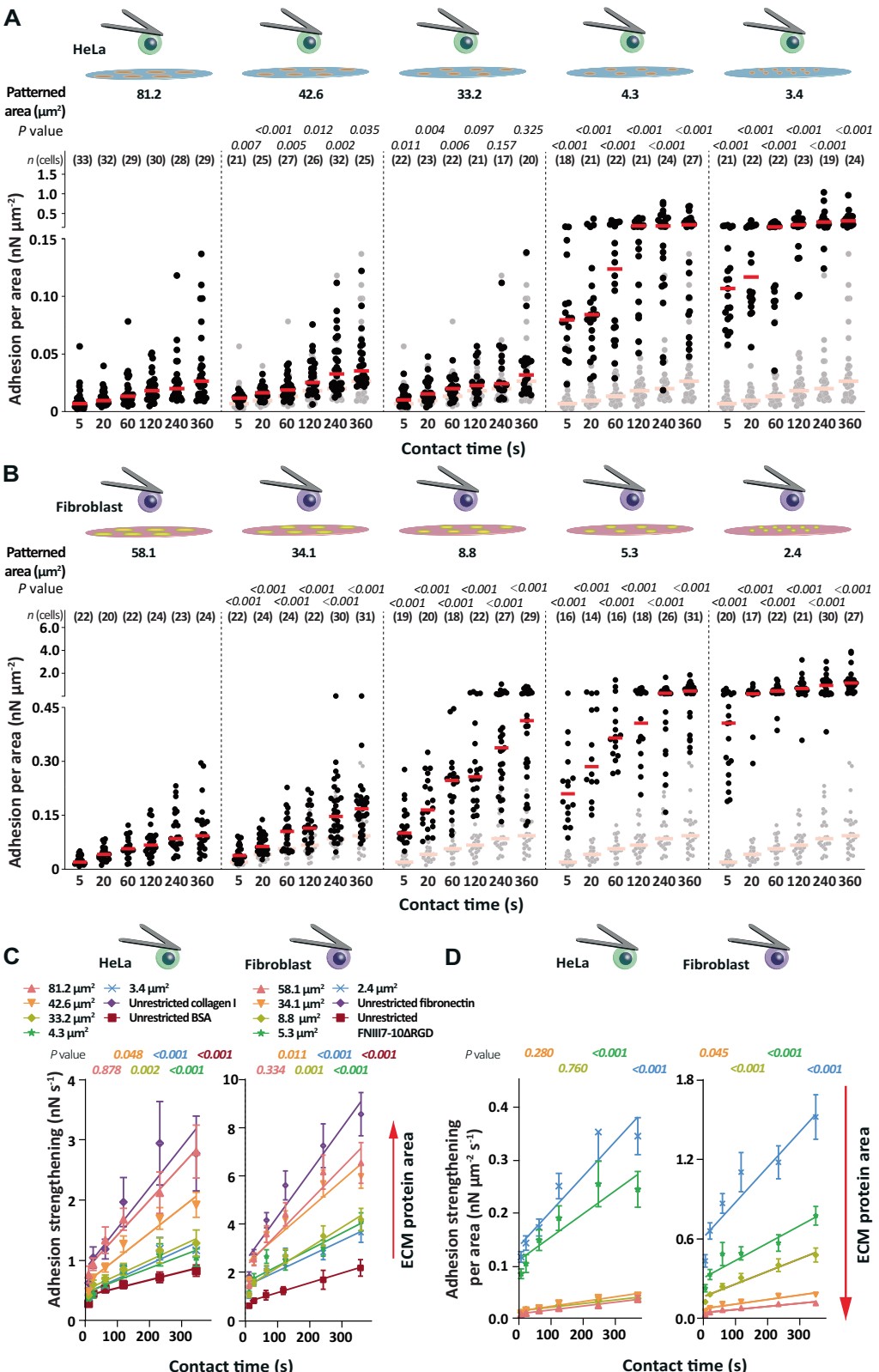

strengthening rate per area to collagen I patterns having areas of ≈ 4.3 µm² and ≈3.4 µm² (Fig. 4D). Similarly, fibroblasts increased the adhesion strengthening rate per substrate area upon decreasing the area of the fibronectin pattern (Fig. 4D).

We next evaluated whether there is a correlation between the circumference of the ECM protein patterns, the area of the ECM protein patterns, and the spatially enhanced adhesion state. We calculated

the circumference of ECM protein patterns using the average area of the ECM protein patterns assuming a perfect circular shape (Supplementary Fig. 7A). The analysis shows that above a circumference-to-area ratio of 1, cells switch to the spatially enhanced adhesion state, whereas below a circumference-to-area ratio of 0.7 cells initiate adhesion canonically (Supplementary Fig. 7B). The finding thus suggests that cells considerably accelerate adhesion strengthening per

**Fig. 4 | Below a certain ECM protein area, HeLa cells and fibroblasts switch into a spatially enhanced adhesion state.** Adhesion force of (**A**) wt HeLa cells and (**B**) wt fibroblasts (data taken from Fig. 3) normalized to the area of the ECM protein pattern (data taken from Fig. 2) at given contact times. Dots represent the adhesion force of single cells per area of ECM protein pattern, red bars median values, and *n* (cells) the number of independent cells tested in at least three independent experiments. Adhesion force per area of (**A**) wt HeLa cells to ≈ 81.2 μm² large printed collagen I patterns or (**B**) wt fibroblasts to ≈ 58.1 μm² large printed fibronectin patterns are shown as reference in semitransparent. *P* values were calculated by two sided Mann–Whitney tests and compare the displayed and reference cell adhesion force. Statistical analysis comparing the potential adhesion forces per area is given

in Supplementary Tables 5, 6. Adhesion strengthening rates of wt HeLa cells or wt fibroblasts as quantified by the slope of a linear regression fit of (**C**) adhesion forces to printed ECM protein pattern or to unrestricted ECM protein substrate (data taken from Fig. 3) and (**D**) adhesion force per area of printed ECM protein pattern for all contact times (data taken from Fig. 4A, B). Dots represent means, error bars SEM, and lines linear regressions. *P* values compare the slopes of data displayed in (**C**) with unrestricted collagen I or fibronectin substrates and in (**D**) with collagen I patterns of ≈81.2 μm² or fibronectin patterns of ≈ 58.1 μm² using one-sided extra sum-of-squares *F* test. Statistical analysis comparing the adhesion strengthening rates is given in Supplementary Tables 9, 10.

---

area to ECM protein patterns having a circumference-to-area ratio >1, indicating that this acceleration is associated to the spatially enhanced adhesion state of the cell.

## Long term adhesion formation depends on the ECM protein pattern area

To address how large integrin-mediated adhesion sites assemble, we attached single, rounded paxillin-GFP expressing HeLa cells or fibroblasts (Supplementary Fig. 8)[8,69] to ConA-coated cantilevers and brought the cells into contact with three different sizes of collagen I patterns (≈ 81.2 μm², ≈ 33.2 μm², and ≈ 3.4 μm²) or fibronectin patterns (≈ 58.1 μm², ≈ 8.8 μm², and ≈ 2.4 μm²). During the contact time of up to ≈ 70 min, we monitored paxillin-GFP every 2.5 min by confocal microscopy. In HeLa cells that were in contact with ≈ 81.2 μm² large collagen I patterns for ≈ 7.5 min, paxillin-GFP started to localize at the periphery of the collagen I pattern. With increasing contact time, the intensity of paxillin-GFP increased both at the periphery and in the center of the pattern. Apparently, more paxillin-GFP localized at the periphery of the collagen I pattern, as found in fixed HeLa cells (Fig. 1F). On the ≈ 33.2 μm² large collagen I pattern, we observed paxillin-GFP to localize in HeLa cells at the periphery of the pattern, but at slower dynamics. After ≈ 15 min contact time, HeLa cells adhering to the ≈ 33.2 μm² large collagen I pattern, started to assemble paxillin-GFP only in some peripheral region of the collagen I pattern. However, the paxillin-GFP covered the entire periphery after ≈ 37.5 min. No clear increase of fluorescent intensity of paxillin-GFP was observed in the center of the pattern. On the smallest collagen I pattern, we observed paxillin-GFP at some peripheral regions after ≈ 7.5 min, while after ≈ 15 min contact time paxillin-GFP covered the entire periphery of the pattern.

In fibroblasts paxillin also localized at the periphery of the fibronectin patterns. On the largest fibronectin pattern of ≈ 58.1 μm², paxillin-GFP started localizing at the periphery at contact times ≈ 7.5 min. Similar to HeLa cells on collagen I patterns, after ≈ 22.5 min, we observed a delayed localization of paxillin-GFP to the intermediate sized fibronectin pattern of ≈ 8.8 μm². On the smallest fibronectin pattern of ≈ 2.4 μm², the fibroblasts started recruiting paxillin already at ≈ 7.5 min contact time.

In summary, we observe paxillin recruitment mainly at the periphery of the ECM protein pattern for HeLa cells adhering to collagen I patterns and for fibroblasts adhering to fibronectin patterns. From our timelapse microscopy experiments we conclude that the dynamics of the paxillin recruitment is ECM protein pattern size depend.

## Kindlin and talin are dispensable for the spatially enhanced adhesion state

Next, we aimed to understand whether integrin activation drives the cellular response to the size of the ECM protein pattern. Thereto, we first incubated wt HeLa cells or wt fibroblasts with Mn²⁺, which induces an extended conformation of integrins[70,71], and quantified cell adhesion to three different sizes of collagen I patterns (≈ 81.2 μm², ≈ 33.2 μm², and ≈ 3.4 μm²) or fibronectin patterns (≈ 58.1 μm², ≈ 8.8 μm², and ≈ 2.4 μm²; Supplementary Fig. 9A, B). The adhesion force of Mn²⁺-treated HeLa cells or fibroblasts to BSA or FNIII7-10ΔRGD was slightly higher than of untreated cells (Supplementary Fig. 10). However, while

Mn²⁺-treated HeLa cells established only marginally higher adhesion force per area to ≈ 81.2 μm² large collagen I patterns, they considerably increased adhesion force per area to smaller collagen I patterns (Fig. 5A). Similarly, Mn²⁺-treated fibroblasts established slightly higher adhesion force per area to ≈ 58.1 μm² large fibronectin patterns and considerably higher adhesion force per area to smaller fibronectin patterns (Fig. 5B). Hence, while the adhesion force of Mn²⁺-treated cells is independent of the ECM pattern area for both cell lines (Supplementary Fig. 9A, B), the adhesion force per ECM protein area considerably increases on small ECM patterns, indicating that integrin activation might be a regulator for sensing the ECM protein area.

To investigate the role of intracellular integrin activators, kindlin and talin, in sensing the ECM protein area, we quantified the adhesion force of talin 1/2-depleted (TKO) and kindlin 1/2-depleted (KKO) HeLa cells or fibroblasts to collagen I or fibronectin patterns (Supplementary Fig. 9C–F). Compared to wt HeLa cells, TKO HeLa cells expressed similar amounts of integrin subunits α1, α2, and β1 on their cell surface, while KKO HeLa cells showed elevated surface expression of the integrin subunits α2 and β1 (Supplementary Fig. 11A). TKO and KKO fibroblasts showed elevated surface expression levels of the integrin subunit α5 and slightly lower surface expression levels of the integrin subunit αv (Supplementary Fig. 11B). Importantly, the unspecific adhesion of the TKO and KKO HeLa cells or TKO and KKO fibroblasts to BSA or FNIII7-10ΔRGD was similar as observed for wt cells (Supplementary Fig. 10). TKO and KKO HeLa cells or TKO and KKO fibroblasts increased the adhesion force to all ECM protein patterns with increasing contact time, which was higher than to BSA or FNIII7-10ΔRGD, verifying integrin specific adhesion (Supplementary Fig. 9C–F). However, the depletion of talin or kindlin drastically reduced the adhesion force of HeLa cells to ≈ 81.2 μm² large collagen I patterns and of fibroblasts to ≈ 58.1 μm² large fibronectin patterns, leading to drastically lower adhesion force per area of TKO and KKO HeLa cells or TKO and KKO fibroblasts (Fig. 5C–F). The adhesion force per area of TKO and KKO HeLa cells and of TKO and KKO fibroblasts to the smaller patterns was not affected by the depletion of talin or kindlin, except for TKO HeLa cells adhering to the smallest collagen I pattern for 120 s and 240 s, for which the adhesion force per ECM protein area was slightly lower. Hence, these results suggest that HeLa cells and fibroblasts can switch into the spatially enhanced adhesion state irrespective the presence or absence of talin or kindlin.

Taken together, the findings suggest that the canonical adhesion initiation of HeLa cells and fibroblasts to ECM proteins depends on kindlin and talin, while both cell lines do not require kindlin and talin to enter and conduct the spatially enhanced adhesion state.

## Cell-specific integrin-actin engagement and integrin-mediated ECM sensing

Next, we aimed to investigate the role of the integrin-actin engagement, which is a major regulator of cell adhesion initiation[18,72,73], on how cells sense the ECM protein pattern area. To this end we first perturbed F-actin using 1 μM latrunculin A (latA) in HeLa cells and fibroblasts. SCFS showed that while latA-treatment did not affect the unspecific adhesion (Supplementary Fig. 10), it diminished the

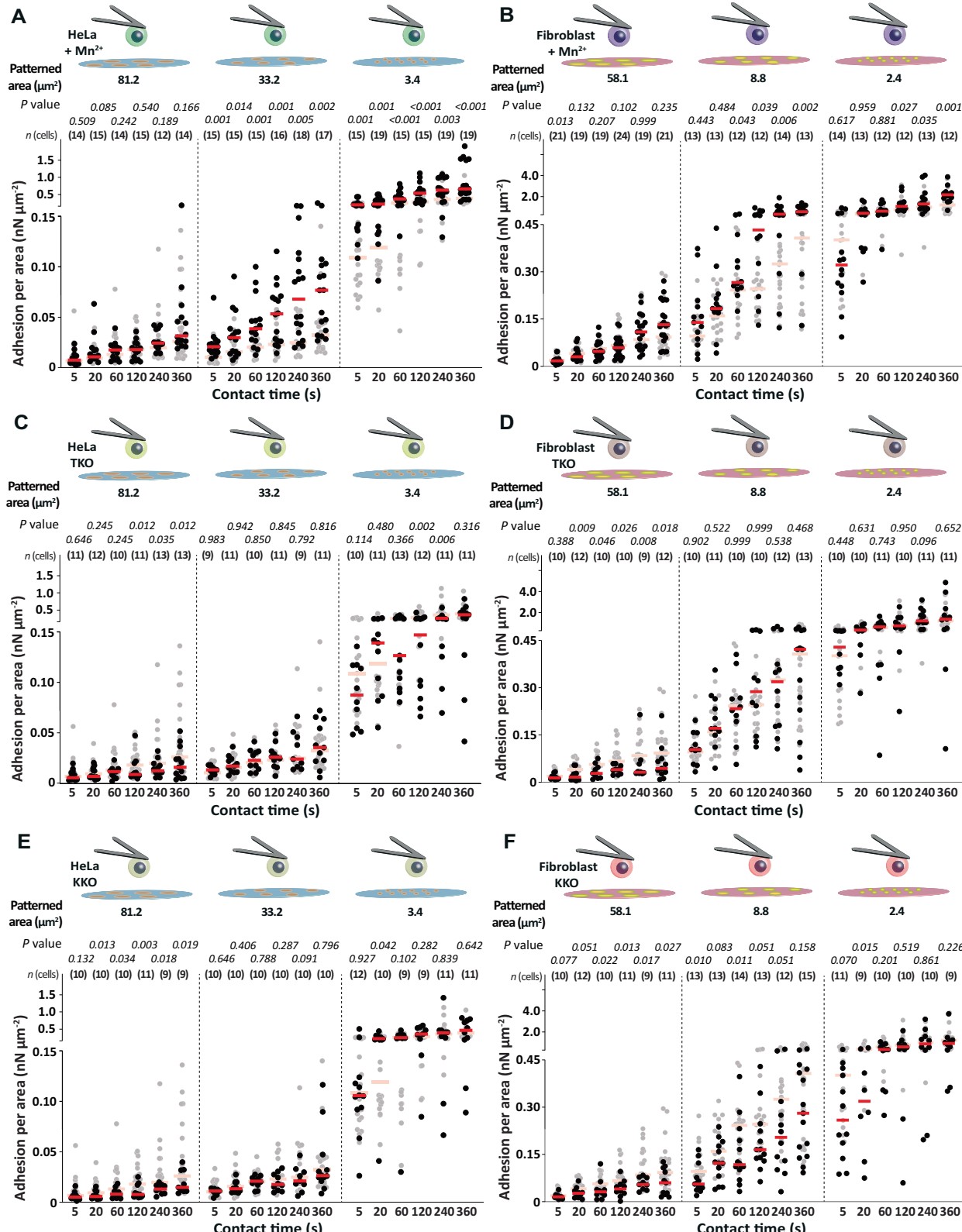

**Fig. 5 | The spatially enhanced cell adhesion state is independent of talin and kindlin.** Adhesion force per area of (**A**) Mn²⁺-treated wt HeLa cells, (**B**) Mn²⁺-treated wt fibroblasts, (**C**) talin-depleted (TKO) HeLa cells, (**D**) TKO fibroblasts, (**E**) kindlin-depleted (KKO) HeLa cells, and (**F**) KKO fibroblasts to different areas of collagen I or fibronectin patterns at given contact times. Dots represent the adhesion force of individual cells, red bars the median, and *n* (cells) the number of individual cells tested in at least three independent experiments. Adhesion force per area of untreated wt HeLa cells or wt fibroblasts in the respective condition are given as reference (semitransparent). *P* values were calculated by two sided Mann–Whitney tests and compare the given adhesion forces with the reference data.

adhesion force and the adhesion force per substrate area of both cell lines to all ECM protein pattern sizes (Supplementary Fig. 12). Hence, an intact actin cytoskeleton is essential to initiate the canonical and the spatially enhanced cell adhesion.

To further test whether actomyosin contractility- and talin-mediated mechanotransduction regulates the ECM protein area dependent cell adhesion, we quantified the adhesion force of myosin II-inhibited wt HeLa cells and wt fibroblasts as well as of TKO HeLa cells or TKO fibroblasts re-expressing the talin1 head domain (TKO + THD HeLa cells or fibroblasts, respectively). Inhibition of myosin II by 20 µM blebbistatin did not affect unspecific adhesion force of HeLa cells or fibroblasts (Supplementary Fig. 10). Myosin II-inhibition lowered the adhesion force of wt HeLa cells to collagen I patterns having an area of $\approx 81.2 \ \mu m^2$ at contact times $\geq 120$ s (Supplementary Fig. 13A). However, myosin II-inhibition did not affect the adhesion force of wt HeLa cells to smaller collagen I patterns. Hence, myosin II-inhibition slightly decreased the adhesion force per ECM protein area of wt HeLa cells to $\approx 81.2 \ \mu m^2$ large collagen I patterns at contact times $\geq 120$ s, while the adhesion force per area was unaffected for smaller collagen I patterns (Fig. 6A). In wt fibroblasts, inhibiting myosin II reduced their adhesion forces to $\approx 58.1 \ \mu m^2$ large fibronectin patterns for contact times $\geq 20$ s and also to $\approx 2.4 \ \mu m^2$ large fibronectin patterns for all contact times except for 5 s and 240 s (Supplementary Fig. 13B). In contrast, the adhesion force of fibroblasts to $\approx 8.8 \ \mu m^2$ large fibronectin patterns remained unchanged. Hence, myosin II-inhibition reduced the adhesion force per area in wt fibroblasts to the largest and smallest fibronectin pattern reduced, while myosin II-inhibition did not affect the adhesion force per area to the intermediate sized fibronectin pattern (Fig. 6B).

TKO + THD HeLa cells and TKO + THD fibroblasts showed similar unspecific adhesion to BSA or FNIII7-10ΔRGD as observed for wt HeLa cells or fibroblasts (Supplementary Fig. 10) and the expression of THD in TKO HeLa cells or HeLa cells did not affect the surface expression of relevant integrins (Supplementary Fig. 11 and Supplementary Fig. 14A, B). Compared to wt HeLa cells, TKO + THD HeLa cells considerably increased adhesion force to $\approx 33.2 \ \mu m^2$ and $\approx 3.4 \ \mu m^2$ large collagen I patterns (Supplementary Fig. 13C). Thus, their adhesion was independent of the overall area of the collagen I patterns. The adhesion force per area of TKO + THD HeLa cells to $\approx 33.2 \ \mu m^2$ and $\approx 3.4 \ \mu m^2$ large collagen I patterns considerably increased, but their adhesion force remained similar to $\approx 81.2 \ \mu m^2$ large collagen I patterns. Hence, expressing talin1 head domain in TKO HeLa cells further strengthened the spatially enhanced adhesion (Fig. 6C). Different to HeLa cells, the adhesion force of TKO + THD fibroblasts to fibronectin patterns was dependent on the pattern area (Supplementary Fig. 13D). Further, compared to wt fibroblasts, TKO + THD fibroblasts generally established lower adhesion force per area to $\approx 58.1 \ \mu m^2$ and $\approx 2.4 \ \mu m^2$ large fibronectin patterns, while the adhesion force per area remained similar to $\approx 8.8 \ \mu m^2$ large fibronectin patterns (Fig. 6D).

In summary, the canonical and the spatially enhanced cell adhesion depend on an intact actin cytoskeleton. However, while the spatially enhanced adhesion of HeLa cells to collagen I is independent of actomyosin contractility within the first 360 s of contact, the spatially enhanced adhesion of fibroblasts to the smallest fibronectin pattern depends on myosin II contractility. Further, the expression of the talin1 head domain in HeLa cells increases the spatially enhanced adhesion, but slightly decreases this adhesion in fibroblasts. These results highlight a cell line and/or integrin specific involvement of actomyosin mediated mechanotransduction and the expression of the talin head domain on the spatially enhanced adhesion.

### Paxillin has a cell line and/or integrin specific effect on the spatially enhanced adhesion

Since the recruitment of paxillin, which is a signaling hub in integrin mediated adhesion[58,59], depends on actomyosin contractility, we investigated the role of paxillin in the cellular response to the ECM protein area by quantifying the adhesion force of paxillin-depleted (PXN KO) HeLa cells or fibroblasts to collagen I or fibronectin patterns (Supplementary Fig. 13E, F). The depletion of paxillin did not affect the unspecific adhesion of HeLa cells and minimally increased the adhesion of fibroblasts (Supplementary Fig. 10). The depletion of paxillin in HeLa cells increased the adhesion force to $\approx 33.2 \ \mu m^2$ and $\approx 3.4 \ \mu m^2$ large collagen I patterns, while it did not alter the adhesion force to $\approx 81.2 \mu m^2$ large collagen I patterns (Supplementary Fig. 13E). Despite of the increased adhesion force, the adhesion of PXN KO HeLa cells remained lower to $\approx 33.2 \ \mu m^2$ and $\approx 3.4 \ \mu m^2$ large collagen I patterns than to $\approx 81.2 \ \mu m^2$ large collagen I patterns. However, PXN KO HeLa cells considerably increased adhesion force per area to $\approx 33.2 \ \mu m^2$ and $\approx 3.4 \ \mu m^2$ large collagen I patterns compared to wt HeLa cells (Fig. 6E). Contrarily, in fibroblasts the depletion of paxillin did not affect the adhesion force nor the adhesion force per area to fibronectin patterns (Fig. 6F, Supplementary Fig. 13F).

In summary, the results indicate that the absence of paxillin has a more pronounced effect on the spatially enhanced adhesion of HeLa cells compared to fibroblasts, thus indicating a cell or integrin type specific effect on adhesion initiation to restricted ECM protein areas. Paxillin modulates HeLa cells to switch to the spatially enhanced adhesion state but not fibroblasts.

## Discussion

Here, we characterized how HeLa cells and fibroblasts initiate and strengthen adhesion to spatially confined ECM protein areas. We patterned surfaces with ECM proteins through µCP, a low cost, simple and versatile method that can be readily applied in cell biological, biophysical, and biomaterial laboratories[33,74–76]. To suppress adhesion of cells to glass, we passivated the non-patterned glass surface with BSA for HeLa cells or FNIII7-10ΔRGD for fibroblasts. This setup reveals that the adhesion force established by HeLa cells to collagen I and by fibroblasts to fibronectin directly correlates with the area of the printed ECM substrate. Thereby, the adhesion force established by HeLa cells to collagen I patterns having areas $\geq 42.6 \ \mu m^2$, is similar to the adhesion force established to spatially unrestricted collagen I. However, HeLa cells establish similarly low adhesion force to collagen I patterns of $\approx 33.2 \ \mu m^2$ and $\approx 3.4 \ \mu m^2$, despite their ten-fold difference in area. Similarly, fibroblasts establish similar adhesion force to fibronectin patterns having areas $\geq 34.1 \ \mu m^2$ and lower adhesion force to fibronectin patterns $\leq 8.8 \ \mu m^2$. Also, the adhesion strengthening of both cell lines shows two states depending on the area of the ECM pattern. These observations suggest that with increasing area of the ECM substrate more integrins can bind ECM ligands until the area of the substrate reaches a certain size after which the cell establishes maximum adhesion. Furthermore, live cell imaging, which shows that paxillin localizes at the circumference of ECM protein patterns, together with the observation that cells increase adhesion force with the pattern size, indicates that increasing the circumference of the ECM protein pattern allows more integrins to bind. Importantly, the results also highlight that HeLa cells and fibroblasts initiate adhesion to decreasing ECM protein areas differently.

Although HeLa cells and fibroblasts establish low adhesion force to patterned substrates below the threshold area, our results show that the adhesion force per area is much higher than in cells adhering to larger ECM protein patterns. Thereby, the number of integrin-ligand bonds established per substrate area increases as the overall area of the ECM substrate decreases. Hence, HeLa cells and fibroblasts switch between two distinct states in response to the overall substrate area: (1) a canonical adhesion state to large ECM protein areas with a low density of integrin-ligand bonds per ECM protein area and (2) a spatially enhanced adhesion state to small ECM protein areas characterized by a high density of ligand-bound integrins per area. For example, HeLa cells establish only <0.7 integrin-ligand bonds per $\mu m^{-2}$ on

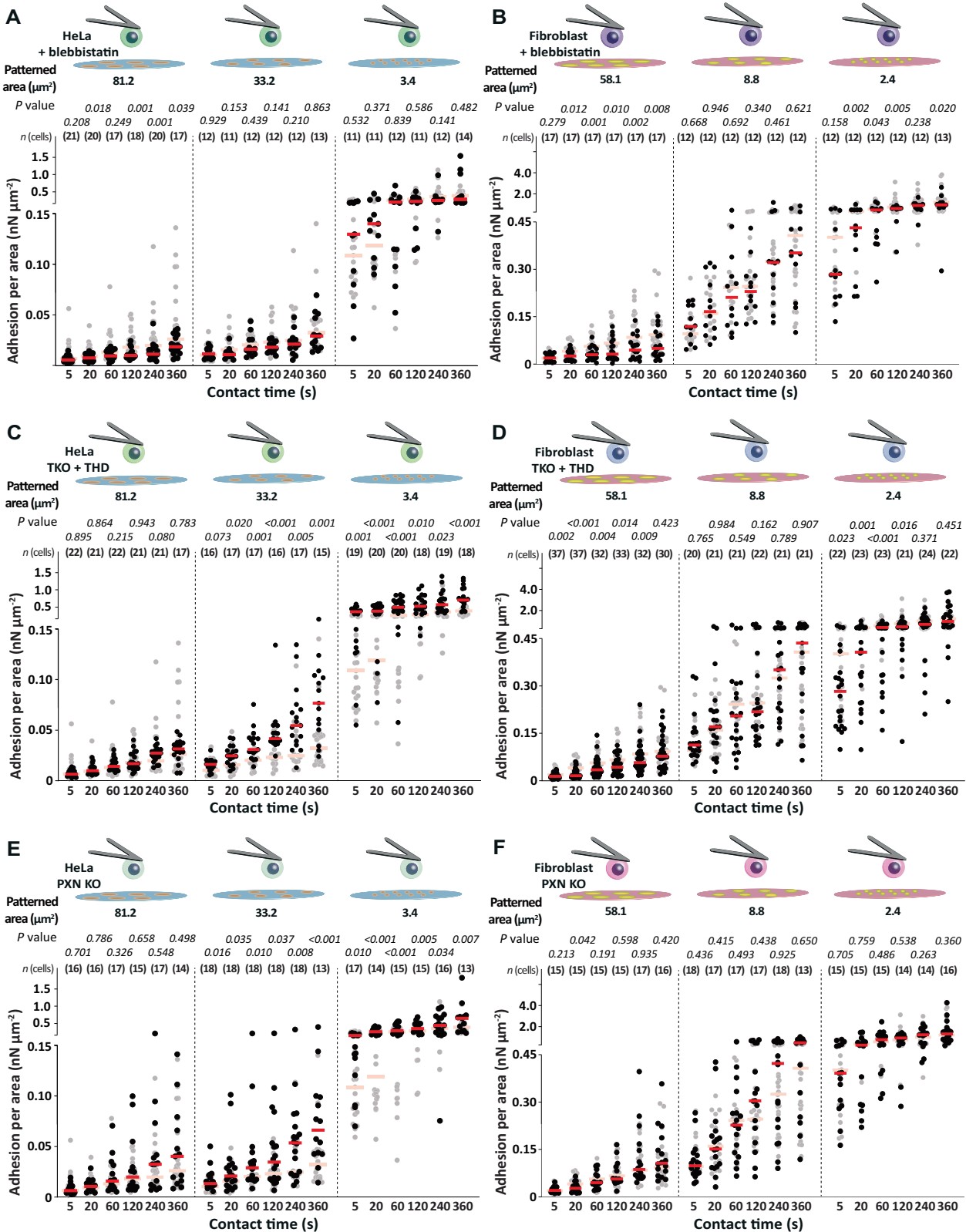

**Fig. 6 | Integrin-actin binding, integrin clustering, and adhesome formation play different roles in the spatially enhanced adhesion state of HeLa cells and fibroblasts.** Adhesion force per area of (**A**) 20 μM blebbistatin-treated HeLa cells, (**B**) 20 μM blebbistatin-treated fibroblasts, (**C**) talin-depleted and talin1-head domain expressing (TKO + THD) HeLa cells and (**D**) TKO + THD fibroblasts, and of (**E**) paxillin-depleted (PXN KO) HeLa cells and (**F**) PXN KO fibroblasts to different areas of printed collagen I or fibronectin patterns at given contact times. Dots represent the adhesion force of individual cells, red bars the median, and *n* (cells) the number of individual cells tested in at least three independent experiments. Adhesion forces per area of untreated wt HeLa cells or wt fibroblasts in the respective condition are given as reference (semitransparent). *P* values were calculated by two sided Mann–Whitney tests and compared the adhesion forces of given data with the reference data.

collagen I patterns ≥ 33.2 μm², but 1–6 integrin-ligand bonds per μm⁻² on collagen I patterns ≤ 4.3 μm². Furthermore, compared to 81.2 μm² large collagen I patterns, the number of integrin-ligand bonds established per area increases ≈ 10-fold on 3.4 μm² large collagen I patterns. Thus, HeLa cells and fibroblasts respond to the size of the ECM protein area by switching between canonical and spatially enhanced adhesion states. Live cell imaging shows that paxillin recruits mostly at the circumference of the ECM protein pattern, which indicates that the circumference-to-area ratio of the ECM protein patterns plays a major role in the cellular switching into the spatially enhanced adhesion state. Indeed, our geometrical analysis reveals that while cells initiate adhesion canonically below a circumference-to-area ratio of 0.7, cells switch to the spatially enhanced adhesion state if this ratio is > 1. The wider range of circumference-to-area ratio in fibronectin patterns (≈ 1.2 to ≈ 2.3) compared to collagen I ( ≈ 1.7 to ≈ 1.9) might explain the more gradual increase in fibroblast adhesion force per substrate area as the fibronectin pattern size decreases. However, we can only speculate about the role of the circumference-to-area ratio in triggering the cells to switch from the canonical to spatially enhanced adhesion strengthening state. One possible explanation would be that ligand-bound integrins are in closer proximity on smaller ECM patterns that show high circumference-to-area ratios, and that this proximity might accelerate integrin clustering during adhesion initiation.

The activation of integrins by $Mn^{2+}$ considerably increases cell adhesion to smaller ECM protein areas and only marginally increases adhesion to larger areas. Hence, by forcing integrins in the extended conformation, cells can increase their adhesion per area on small ECM patterns indicating that integrin activation further strengthens the spatially enhanced adhesion. The depletion of talin and kindlin, which are required for intracellular integrin activation in cell adhesion to unrestricted ECM proteins, results in the loss of adhesion adjustment in response to the ECM protein area by only decreasing the adhesion of cells to large ECM protein areas. Interestingly, our results indicate that cell adhesion initiation to smaller ECM protein patterns (collagen I patterns ≤ 33.2 μm², fibronectin patterns ≤ 8.8 μm²) is mostly independent of kindlin or talin. Hence, the spatially enhanced adhesion state appears to be fundamentally different from cells adhering in the canonical adhesion state to large or unrestricted areas of ECM proteins. Since we show that integrins detach at similar forces from the ECM ligand, independent of the area of the ECM protein, this finding further indicates that the spatially enhanced adhesion is dominated by intracellular regulation.

We also find that the adhesion established by HeLa cells expressing only the talin1 head domain is independent of the collagen I pattern area. Whereas the expression of the talin1 head domain strengthens the spatially enhanced cell adhesion, the adhesion to large collagen I areas remains unaltered. The similarity of the adhesion response of talin1 head domain expressing TKO HeLa cells and of HeLa cells having $Mn^{2+}$-activated integrins suggest that the expression of the talin1 head domain, which is constitutively active in binding to integrins, hyperactivates integrins[77]. This finding suggests that the spatially enhanced adhesion state can be strengthened by talin activation and recruitment to integrin. Further, these results together with the independence of the spatially enhanced adhesion initiation on myosin II activity indicate that the spatially enhanced adhesion state of HeLa cells does not require integrins to be engaged to the actomyosin cortex and hence does not depend on mechanotransduction. Contrary, the adhesion of fibroblasts expressing only the talin1 head domain depends on the fibronectin area. Additionally, the spatially enhanced adhesion of fibroblasts to the smallest fibronectin pattern depends on myosin II-mediated actin contractility. This indicates cell or integrin type specific contributions of the intracellular recruitment of talin and actomyosin contractility to integrins in sensing the size of ECM protein area and the regulation of the spatially enhanced adhesion. We also report a cell or integrin type specific function of paxillin

in initiating adhesion. While paxillin depleted HeLa cells increase adhesion to small collagen I patterns, which further increases the spatially enhanced adhesion, paxillin depletion leaves the fibroblast adhesion unaltered. The results, thus, indicate that actomyosin contractility and adhesome formation play cell and/or integrin type specific roles in sensing the ECM protein area and triggering the cellular response to switch the adhesive state.

Many researchers expose cells to restricted patterns to control cell adhesion, polarization, and migration[6,38,78–80]. The results presented here show that cells can respond to the area and/or circumference of confined substrates to which they respond through regulating adhesion from a normal, canonical adhesion to a spatially enhanced adhesion state. Unraveling this cellular mechanism of response to the ECM area offers new strategies to better understand cellular responses in confined environments such as those provided in vivo by the ECM and to guide cell adhesion initiation and strengthening through the design of patterned substrates. However, the observations made here also open new questions, such as how do cells respond to complex ECM architectures, which sometimes only expose fractions of ECM components? Another key question arising would be whether, in such case, cells respond stronger to smaller ECM areas compared to larger ones and, thus whether less ECM could trigger stronger cellular responses. Such dependencies may become particularly important in ECMs being composed of different ratios of ECM proteins.

## Methods
### PDMS stamps for μCP
Arrays of micropillars having diameters of 10 μm, 8 μm, 5 μm, 3 μm, and 2 μm for μCP were formed through replication molding using Si masters. The Si master was produced by photolithography and ion beam processes. The micropillar pattern was transferred to a UV light sensitive photoresist on a silicon substrate by photolithography using a glass mask with the required pattern. After layer development, holes were etched into the Si master to the desired depth using ion beam. Finally, the resist was removed with plasma asher. Poly(dimethyl) siloxane (PDMS) elastomer (Sylgard 184, Dow corning) was mixed in a 10:1 (wt/wt) ratio of prepolymer and curing agent. PDMS was degassed in vacuum for 30 min. The PDMS was casted on ethanol washed silicon wafers and cured at 100 °C for 1 h. Peeled-off PDMS pillar stamps were sonicated in 70% ethanol followed by deionized water and dried with pressurized air.

### Microcontact printing (μCP) of ECM proteins
Collagen I (PureCol® Type I Collagen, 5005, Advanced BioMatrix) or fibronectin (Fibronectin, Bovine Plasma, 341631, Sigma Aldrich) were fluorescently labeled by incubating 50 μg ml⁻¹ protein solution (in PBS) with Alexa fluor 555 (Alexa Fluor™ 555 NHS Ester, A20009, Thermo Fisher Scientific) in a 1:100 (vol/vol) dilution for 10 min at RT. The PDMS stamps were incubated with 50 μg ml⁻¹ Alexa fluor 555-labeled collagen I or fibronectin for 30 min at 25 °C. Inked PDMS stamps were washed twice with PBS and deionized water and dried with pressurized air for 30 s. Subsequently, the PDMS stamp was placed on glass surfaces of a Petri dish (fluoro dish, FD35-100, World Precision Instruments) for 5 min to allow complete transfer of the protein. For passivation, the glass surfaces were incubated with fluorescein isothiocyanate (FITC) conjugate Bovine Serum Albumin (BSA, A9771-50MG, Sigma Aldrich) or Alexa fluor 488 (Alexa fluor™ 488 NHS Ester, A20000, Thermo Fisher Scientific)-labeled FNIII7-10ΔRGD (50 μg ml⁻¹) at 4 °C overnight. Prior to using the dishes in experiments, they were washed with PBS to remove residual proteins.

### Characterization of printed ECM protein patterns
Printed collagen I or fibronectin patterns were characterized by AFM (NanoWizard II, JPK) imaging in contact mode at RT using triangular cantilevers with nominal spring constants of 0.35 N m⁻¹ (SNL-10,

Bruker). Patterned surfaces were first imaged 50 μm x 50 μm for an overview image at 256 x 256 pixels at a line rate of 1.8 Hz. Subsequently, single printed ECM protein patterns were imaged at 256 x 256 pixels. The AFM images were collected from at least five different patterned surfaces with more than 40 printed patterns in total. Gains and cantilever deflection were adjusted during the scans to acquire best possible images at the lowest possible imaging force applied.

Height profiles of printed patterns were obtained from cross sections using the AFM inbuild data processing software (JPK data processing 7.0). To measure the diameter of the patterns, the full width at half average height was measured for individual patterns. To quantify the area of the patterns, a binary height mask of single patterns was created at the half average height using the AFM data processing software. Binary images were imported to ImageJ, transformed into gray scale images and the area of the patterns was quantified using the analyze particles command.

## Cell line engineering

CRISPR/Cas9-mediated gene knockout targeting paxillin (PXN) was performed in wild type (wt) HeLa (Kyoto) cells and mouse embryonic kidney fibroblasts. Utilizing the online single-guide RNA (sgRNA) design tool CHOPCHOP[81], two sgRNAs were generated for each cell type to ensure complete PXN depletion. The sgRNAs sequences for HeLa cells, based on the human genome database (GRCh38/hg38), were 'GGGGTCGCCACAGTCGCCAA' and 'GATCCCGGAACTTCTTCGAGC', while for fibroblasts, based on the mouse genome database (mm10/GRCm38), the sequences were 'GTAAGGTCGTGACCGCCATG' and 'GAGGCGCACACGAAATGCTCG'. Primers encoding the sgRNAs were synthesized, annealed and ligated into fluorescent protein-tagged Cas9 expression plasmids following a protocol outlined earlier[82]. For each knockout experiment, the two sgRNAs were individually cloned into green fluorescent protein-tagged Cas9 plasmid (pSpCas9-2A-GFP)[82] and blue fluorescent protein-tagged Cas9 plasmid (pSpCas9-2A-BFP)[83]. The plasmids were transiently co-transfected into HeLa cells or fibroblasts using lipofectamine 2000 (11668019, Invitrogen) according to the manufacturer instruction. After 48 h of transfection, HeLa cells and fibroblasts were detached, pelleted, and resuspended in PBS at a concentration of $10^6$ cells ml$^{-1}$. GFP and BFP double-positive HeLa cells and fibroblasts were sorted into 96-well plates as single-cell colonies using a fluorescence activated cell sorter (BD FACSMelody, USA). PXN depletion in single cell colonies was verified using Western blot analysis (Supplementary Fig. 14C). For that, cells were lysed using RIPA lysis and extraction buffer (89900, Thermo Fisher Scientific) and the supernatant was subjected to electrophoresis on a 4–12% Bis-Tris gel (NuPAGE 4 to 12%, Bis-Tris, 1.0 mm, Mini, NP0322BOX, Invitrogen) and subsequently transferred onto nitrocellulose membranes (Amersham Protran 0.45 μm NC, 10600002, GE Healthcare). PXN was detected using rabbit anti-PXN (1:2000, anti-paxillin antibody [Y113], ab32084, Abcam). PXN KO HeLa cells and PXN KO fibroblasts used for SCFS experiments were expanded from a verified single cell clone (Supplementary Fig. 14C). To verify the talin expression level in wt fibroblasts, TKO fibroblasts and TKO + THD fibroblasts, talin was detected using anti-talin 1,2 (1:100, monoclonal anti-talin antibody produced in mouse, T3287, Sigma), anti-talin1 (1:500, anti-talin 1 antibody [97H6], ab108480, Abcam), and anti-talin2 antibodies (1:500, anti-talin 2 antibody [68E7], ab105458, Abcam). Rabbit anti-GAPDH antibody (1:2000, anti-GAPDH antibody, ab9485-100UG, Abcam) was used as a loading control for all blots. To verify the talin expression level in wt HeLa cells, TKO HeLa cells and TKO + THD HeLa cells, talin was detected using anti-talin head (1:100, talin antibody TA205, MCA725G, Bio-rad). The secondary antibody used was goat anti-rabbit antibody (1:2000, goat anti-rabbit IgG H&L (HRP), ab205718-500UG, Abcam) and goat anti-mouse antibody (1:2000, Goat Anti-Mouse IgG H&L (HRP), ab205719-500UG, Abcam). Chemiluminescence detection was performed using a FUSION PULSE TS imaging system (Vilber).

## Cell culture

Wild type HeLa (Kyoto) (kind gift from A. Hyman, MPI Molecular Cell Biology and Genetics, Germany), TKO HeLa cells[49], TKO + THD HeLa cells[49], KKO HeLa cells[49], PXN KO HeLa cells, wild type mouse embryonic kidney fibroblasts[39], TKO fibroblasts[19], TKO + THD fibroblasts (kind gift from C. Grashoff, University of Munster, Germany), KKO fibroblasts[19] and PXN KO fibroblasts were cultured in Dulbecco's modified eagle medium (DMEM, 31966047, Thermo Fisher Scientific), supplemented with 10% (vol/vol) fetal bovine serum (FBS, F9665, Sigma Aldrich), 100 U ml$^{-1}$ penicillin and 100 μg ml$^{-1}$ streptomycin (15140122, Thermo Fisher Scientific).

## Single-cell force spectroscopy (SCFS)

Tipless AFM cantilevers with a nominal spring constant of 0.06 N m$^{-1}$ (NP-O, Bruker) were plasma cleaned (PDC-32G, Harrick Plasma) for 5 min and incubated overnight with 2 mg ml$^{-1}$ concanavalin A (ConA, C2010-100MG, Sigma Aldrich) in PBS at 4 °C. Cells were grown in 12-well plates to a maximal confluency of ≈ 80%. Before experiments, cells were serum-starved for at least 1 h before SCFS. Then cells were washed with PBS and detached with 200 μl of 0.25% (w/v) trypsin/EDTA (25200072, Thermo Fisher Scientific) for 2 min at 37 °C. Detached cells were suspended in 1% (vol/vol) FCS containing SCFS medium (DMEM, 12800017, Thermo Fisher Scientific) supplemented with 20 mM HEPES. Cells were pelleted and resuspended in 200 μl FCS-free SCFS media. After detachment, cells were allowed to recover for 30 min from trypsin/EDTA treatment in SCFS medium at 37 °C (Ref. 63). For SCFS in the presence of Mn$^{2+}$, blebbistatin, or latrunculin A, suspended cells were incubated with 0.5 mM MnCl$_2$ (Manganese(II) chloride solution, M1787, Sigma-Aldrich), 20 μM blebbistatin (203390, Sigma-Aldrich), or 1 μM latrunculin A (L5163, Sigma-Aldrich) in SCFS medium for at least 30 min and the chemicals were present throughout the experiments.

SCFS was performed using an AFM-based CellHesion 200 or an AFM (NanoWizard II) equipped with a CellHesion-module (all JPK instruments) mounted on an inverted microscope (AxioObserver, Zeiss). The ambient temperature was maintained at 37 °C by a PetriDish-Heater (JPK instruments). The exact spring constant of each used cantilever was calibrated prior to experiments using the thermal noise method. To attach a single cell to the ConA-coated cantilever, suspended single cells were pipetted onto BSA or FNIII7-10ΔRGD-coated areas of Petri dishes. The cantilever was lowered onto a single cell with 10 μm s$^{-1}$ until detecting a contact force of 5 nN. After 5 s contact time the cantilever was retracted at 10 μm s$^{-1}$ by > 90 μm to fully separate cell and substrate. Cells were then incubated for 10 min on the cantilever to ensure firm binding. Cells of similar size and morphology were attached to cantilevers to minimize possible variations. The cell morphology was monitored throughout the experiment using optical microscopy to ensure that only cells having round morphologies were characterized.

Cell adhesion forces were quantified by approaching single cells to the patterned protein substrate at 5 μm s$^{-1}$ until recording a contact force of 2 nN. The cantilever was maintained at constant height for contact times of 5, 20, 60, 120, 240, or 360 s. Thereafter, the cantilever-bound cell was retracted from the substrate at 5 μm s$^{-1}$ for 100 μm until cell and substrate were fully separated. After the experimental cycle, cells were allowed to recover from adhesion measurement for the time of contact time before measuring the adhesion force for a different contact time. The order of contact times was randomized for each cell to exclude potential memory effects of the cells on experimental sequences. The printed protein pattern on the substrate was altered after every adhesion force measurement. Adhesion forces of cantilever bound cells were quantified for all contact times unless morphological changes, such as cell spreading or cell division, were detected. Adhesion forces were determined from retraction force-distance curves after drift-and baseline correction using JPK data analysis software (JPK

data processing 7.0). To correct adhesion forces on ECM protein patterns, median value of wt HeLa cells to unrestricted BSA and wt fibroblasts to unrestricted FNIII7-10ΔRGD at respective contact time is given as reference. The correction analysis subtracted these reference values from adhesion forces measured on ECM protein patterns, with any results below zero being cut off. Due to the similar adhesion force established by wt, TKO, KKO, blebbistatin-treated, latrunculin A-treated, TKO + THD, PXN KO HeLa cells to unrestricted BSA and wt, TKO, KKO, blebbistatin-treated, latrunculin A-treated, TKO + THD, PXN KO fibroblasts to unrestricted FNIII7-10ΔRGD respectively, only adhesion force of wt cells to protein patterns is analyzed. Adhesion force strengthening was determined as the slope of linear fits to all adhesion forces and contact times (PRISM 8.2.1).

### Analyzing single rupture and tether forces from SCFS data
Single rupture forces and tether forces from force-distance curves recorded by SCFS across all contact times for each pattern size were pooled and analyzed using the AFM data analysis software (JPK data processing 7.0)[42]. Rupture events were identified by the non-linear slope before of the force jump, while tether events were identified by the force plateaus having a maximum tilt of 10° before of the force jump. A cutoff of 24.60 pN, which corresponds to twice the noise of our SCFS experiment, was applied before analyzing rupture and tether forces.

### Confocal microscopy
Proteins used for patterning were labeled with Alexa fluor 555 according to the instructions and imagined using confocal microscopy (LSM700 equipped with EC Epiplan 10x/0.2 M27 objective, Zeiss) to detect the patterned ECM protein substrates. To detect the single cell attached to the cantilever and contacted with patterned protein substrate, cells were incubated in DMEM (31966047, Thermo Fisher Scientific) and 100 U ml$^{-1}$ penicillin and 100 µg ml$^{-1}$ streptomycin with CellTracker (CellTracker™ Fluorescent Probes, C2925, Thermo Fisher Scientific) with 1:1000 dilution at 37 °C for 30 min. After incubation, cells were washed with PBS to remove the staining solution and imaged with confocal microscopy (LSM700 with a LCI Plan-Neofluar 63x/1.3 Imm Korr DIC objective, Zeiss). To image adhesion sites formed on patterned ECM protein substrates, wt HeLa cells and wt fibroblasts are seeded onto printed Alexa fluor 405 (A30000, Thermo Fisher Scientific)-labeled collagen I or fibronectin patterns for 60 min. Afterwards, weakly attached HeLa cells or fibroblasts were removed by gentle washing with PBS. Then cells were fixed using 4% (v/v) paraformaldehyde (28908, Thermo Fisher Scientific) for 15 min at room temperature (RT). 0.2% (v/v) Triton X-100 (T9284, Sigma Aldrich) was used to permeabilize cells for 5 min at 25 °C. PBS with 0.1% (v/v) Tween-20 (P2287, Sigma Aldrich) and 3% (w/v) BSA (A2153, Sigma Aldrich) was used to block the sample for 30 min at RT. Samples were incubated with primary antibody Rabbit anti paxillin (1:100; Anti-Paxillin antibody [Y113], ab32084; Abcam) in blocking solution at 4 °C overnight. Afterwards samples were incubated with donkey anti rabbit Alexa fluor 488 (1:200; ab150073; Abcam) and rhodamine phalloidin (1:500; R415, Thermo Fisher Scientific) in blocking solution for 1 h at RT. Samples were imaged and analyzed with laser scanning confocal microscopy (Zeiss LSM980 IR with a C-Apochromat 40x/ 1.20 W Korr). To conduct timelapse confocal microscopy images, wt HeLa cells and fibroblasts overexpressing paxillin-GFP were maintained at SCFS for 70 min. During the contact time paxillin-GFP localization was monitored every 2.5 min using laser scanning confocal microscopy (LSM700 with a LCI Plan-Neofluar 63x/1.3 Imm Korr DIC objective, Zeiss). Laser intensities and gains were optimized for each cell prior to experiments. After every experiment, the cantilever and the cantilever-bound cell were exchanged. Images were processed by Zeiss Zen 3.9.

### Flow cytometry
Cells were serum starved for at least 1 h, trypsinized, washed with PBS and 6 x 10$^5$ cells were resuspended in 500 µl PBS. Unlabeled wt HeLa cells and fibroblasts were analyzed as a negative control. TKO, KKO, TKO + THD, PXN KO or wt HeLa cells were incubated with antibody against integrin α1 (1:100; FITC anti-integrin α1, ab34176, Abcam), α2 (1:200; FITC anti-human CD49b, 359305, Biolegend), β1 (1:40; anti-integrin β1, 11-0291-82, Thermo Fisher Scientific) in PBS on ice for 1 h. TKO, KKO, TKO + THD, PXN KO or wt fibroblasts were incubated with antibody against integrin α5 (1:100; anti-integrin α5, ab25189, Abcam), αV (1:100; anti-integrin αV, 12-0512-82, Thermo Fisher Scientific), β1 (1:40; anti-integrin β1, 11-0291-82, Thermo Fisher Scientific) in PBS on ice for 30 min. Cells incubated with antibodies against integrin α5 were washed with PBS twice and incubated with streptavidin-phycoerythrin (1:1000; ab239759, Abcam) in PBS on ice for 30 min. Following antibody incubation cells were washed twice with cold PBS and finally resuspended in 250 µl PBS and kept on ice. Fluorescence intensities of single cells were analyzed using a flow cytometer (Fortessa, BD Bioscience). Laser intensities were optimized for each experiment and maintained constant for conditions to be compared. Flow cytometry data was gated according to forward and side scatter to exclude debris and doublets (Supplementary Fig. 15). FlowJo V10 software was used to analyze flow cytometry data.

### Statistics & reproducibility
All data presented were acquired in at least three independent experiments. No statistical method was used to predetermine sample size. No data were excluded from the analyses. The experiments were not randomized. The investigators were not blinded to allocation during experiments and outcome assessment. Statistical tests, such as indicated in the figure legends, were performed using Prism and two-tailed unpaired, non-parametric Mann–Whitney tests. To statistically compare adhesion strengthening under different conditions, a linear regression analysis of the adhesion force recorded for all contact times was performed. The one-sided extra sum-of-squares $F$ tests were used to statistically test the adhesion strengthening, rupture and tether force under different conditions.

### Reporting summary
Further information on research design is available in the Nature Portfolio Reporting Summary linked to this article.

## Data availability
The data generated in this study have been deposited in the ETH research data collection and is available under: https://doi.org/10.3929/ethz-b-000735457. Source data are provided with this paper.

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

## Acknowledgements

We thank J. Huang for providing competent cells, G. Fläschner and P. Rimpf for providing silicon masters, M. Huber for support in SCFS, cell culture and western blot. We thank the single-cell facility of the Department of Biosystems Science and Engineering, ETH Zürich for their support. This work was supported by the Swiss National Science Foundation (SNS, grants no. 31003A_182587/1 to DJM and 310030_215690/1 to DJM) and the Swiss National Science Foundation as part of the NCCR Molecular Systems Engineering (Grant 51NF40-205608 to DJM).

## Author contributions

X.W., N.S. and D.J.M. designed the experiments and wrote the paper. X.W. performed and analyzed most experiments. N.S. helped with experimental set up and data analysis. P.W. and L.Z. helped with cell line engineering. T. X. helped with PXN KO fibroblast cell line sorting. T.X. and H.Y. helped with PXN KO fibroblast western blotting. S. A. helped with microcontact printing. U.S. helped with PXN KO HeLa cell line sorting and western blotting. All authors discussed the experiments, read, and approved the manuscript.

## Competing interests

The authors declare no competing interests.
