## [Transparent Peer Review file · Nature Communications]

Mammalian cells measure the extracellular matrix area and respond through switching the adhesion state

Corresponding Author: Professor Daniel Müller

Version 1:

Reviewer comments:

Reviewer #1

(Remarks to the Author)

In their manuscript, Wang et al. set out to study cell adhesion to spatially confined ECM ligands, with focus on early stages of cell adhesion formation as quantitated by single cell force spectroscopy. To spatially confine ECM ligands, they combine the AFM assay with micropatterned glass surfaces functionalised with collagen and fibronectin. Thereby patterns of increasing sizes are created, ranging from 2.4 μm^2 to 81 μm^2 , onto which single cells, immobilised on an AFM cantilever, are pushed for a defined contact time. As expected, they find that with increasing contact time and pattern size, overall cell adhesion forces increase. When they, however, normalise the overall adhesion forces to the respective contact areas, they find surprisingly over proportionally enlarged adhesion forces per area for the small patterns. They conclude from this that the studied cell types (HeLa cells and fibroblasts) can actively measure the size of the substrate area and switch to a “spatially enhanced adhesion state”. After conducting a set of experiments using integrin activating agents, talin and kindlin knockout cells, as well as cells overexpressing the talin head domain (supposedly increasing integrin activation), they come to the conclusion that cells enter this “spatially enhanced adhesion state” independently of talin or kindlin, while integrin activation, clustering and mechano-transduction pathways might be required, albeit in a cell-type dependent manner. In their manuscript, the authors present an extensive and overly interesting dataset that might point at a novel mechanism by which cells adjust their adhesion to spatially restricted ECM ligands. However, some of the main conclusions appear not fully supported by the presented data. Overly, the underlying mechanisms remain rather vaguely described. There are multiple questions and points that should be addressed for clarity.

Major points:

The authors claim in the abstract that mechanistic insights were obtained (abstract line 31/32). They conclude “that the actin cytoskeleton along with integrin clustering and adhesome formation plays a cell and/or integrin type specific role in sensing ECM, responding, and state switching to adhere to the ECM protein area.”(line 374).

These statements, however, seem not well supported by the shown data. There were no experiments conducted where f-actin or actomyosin contractility was perturbed. For instance, experiments interfering with actomyosin contractility, as performed by Changede and al ([https://www.cell.com/developmental-cell/fulltext/S1534-5807\(15\)00715-7#mmc1](https://www.cell.com/developmental-cell/fulltext/S1534-5807(15)00715-7#mmc1)) could be performed to support this statement.

If integrin clustering was involved, this should also be visible from analysing the single rupture events (especially at prolonged contact times). However, the presented data are not explained well enough (e.g. for which contact time were these rupture forces analysed in Supplementary figure 5?) to understand this. In addition, analysis of the single unbinding events at prolonged contact times might help to understand whether clusters had formed, since this should result in an increase of the smallest rupture events. It would be interesting to compare whether an eventual increase of individual rupture events already occurs at earlier time points on smaller areas. In addition, the formation of larger adhesion sites (e.g. fluorescently tagged paxillin or vinculin recruitment) could be supported by imaging data, e.g. combining AFM and confocal or TIRF microscopy. Alternatively, they could perform live imaging after adding the cells over the micropatterned surface and imaging cells as soon as they start to attach.

As it appears, the number of bound integrins is estimated from the single rupture forces determined (dataset in Supplementary Fig 5) and the overall adhesion force. This appears over-simplified, since the steps as shown in the force curve example (Supplementary Figure 5) might not be simply add up to the maximal force due to spring-like force loadings in between. In addition, the tether forces would contribute significantly to the overall detachment forces. Since the authors also

analyse the single rupture and tether events, it appears more appropriate to present the actual number of rupture events per curve (then also rupture and tethers could be distinguished).

Given the not insignificant contribution of unspecific adhesion, it appears logic to correct for the non-specific reference (as done in Suppl fig. 4). From the methods it appears that a constant median value on passivated surfaces was simply subtracted. Would it not be more accurate to subtract proportional to the area of unspecific coating that the cell contacts on each of the patterns (which becomes relatively larger for the smaller patterns)? There could potentially arise a bias for larger patterns that were mostly in contact with the ECM ligand but not the surrounding passivated surface. Is this effect taken into account by the explained method?

Are there any geometric considerations that should be taken into account? E.g., the process of detaching/"peeling off" the cell from the substrate could be different for larger and smaller areas. Since the cell contact area probably "shrinks" from the outside to the inside during the detachment process, would the results be different if the adhesion energy and not the overall detachment force would be plotted? In any case, it would be relevant to plot the adhesion work/energy over time.

Was the overall surface contact area also quantitatively measured for the different patterns, e.g. using a combined confocal AFM setup? It might be possible that the cell shows distinct spreading on the different substrates at longer contact times. This might affect overall adhesion forces, since the cells also adhere to some extent with the surrounding area (e.g. "unspecific adhesion to BSA"). From Supplementary figure 1E it is not clear whether the graph presents different patterns that are pooled?

How were the paxillin and talin knockout cells created? And, were they thoroughly characterised with regards to integrin expression? If they were expanded from single cell clones, there might be intrinsic differences in integrin expression and/or surface levels. This should be checked, since many conclusions rely on a direct comparison of WT and KO cell lines.

The TKO results appear rather difficult to be interpreted, since they are sometimes inconsistent between contact times. The authors state that (Line 260) "the adhesion per area was not affected by TKO on smaller patterns". This seems not fully supported for HeLa cells by the results shown in Fig. 5C showing for some time points significant changes for both, larger and smaller patterns.

There are also multiple rather strong statements that seem not to be sufficiently supported by the presented data. To give two examples:

248/249: "... indicates that integrin activation is an essential regulator for sensing the ECM protein area."

Similarly, it is stated in 341/342: "Hela cells and fibroblasts can measure the size of the substrate area and in response switch between canonical and spatially enhanced adhesion states"

While the results show that cell adhesion is proportionally higher on smaller substrates, and that Mn treatment enhances this effect, it is not clear whether that has to do with active sensing of the ECM contact area.

Which effect would lateral diffusion of integrins into the contact zone have, especially at longer contact times. Given that on small patterns the length of the contact zone edge is relatively enhanced, could this also contribute to more recruitment of receptors from nearby regions? It would be great to see this discussed. Also, it might be possible to conduct experiments, e.g. based on confocal/TIRF imaging as mentioned above, or experiments with truncated integrins to explore this effect.

Minor:

It would be great to see higher resolution images for figure 1E to be able to properly see the adhered cells.

The style the manuscript is written it is somewhat difficult to follow in some parts, especially the part describing the results with talin and paxillin KO cells reads very cumbersome. It would also be helpful, if the motivation for the following sections (e.g. paxillin KO) would be better explained. Also, the abstract could be sharper, e.g. by mentioning a hypothesis or knowledge gap that the paper is addressing.

In the figures, a plethora of data acquired under different conditions (multiple time points, different reference and multiple patterns) is shown. The visualisation of the plots in Fig. 3-6 is sometimes difficult to read. While it nice to have the partly transparent reference values in the background, it would be good to see the medians in the foreground to compare them to the other medians (some are completely invisible). Also, there are two distinct populations in the normalised data. Did the authors try to perform a cluster analysis and do statistics on the percentage of cells in the two described "states"?

In Supplementary Figure 5, the mean rupture force is given to describe the distributions. Since most distributions show a truncated Gaussian distribution. Since they can be robustly fit by a Gaussian, it would make sense to also present the modes. In addition, the axis is labelled with counts, so it should probably read relative counts, counts per x or frequency.

The references are missing. Either this is an issue with PDF version or the authors have submitted an incomplete reference list (references are only till 37 whereas cited number in text goes up to 70).

A reference for role of Mn²⁺ in integrin activation should be added.

The authors show a Western blot for the paxillin KO cell lines to show absence of paxillin. Why not show for the talin and kindlin KO cells as well or refer to published data? Also, how do the integrin and talin levels of TKO+THD compare to wildtype cells used as a reference?

Reviewer #2

(Remarks to the Author)

Reviewer #3

(Remarks to the Author)

This is an interesting biophysical approach to looking at how cells respond to small areas of ECM. As someone who has looked at cells adhering to materials and ECM for decades, I was surprised that fibroblasts don't stick to collagen and HeLa don't stick to fibronectin - I looked at the literature as I was unsure and there are reports of there of both cell types adhering to both ECMs? Why also the differential blocking regime - BSA and RGD depleted FN? I am sure the authors can explain this convincingly.

I am not a biophysicist, I am a bioengineer/cell biologist - I need to have more information about rupture force. How does it show single integrin events for a whole cell being retracted from the ECM patterns?

The authors show that HeLa and fibroblasts increase adhesion, rupture force and adhesion strengthening to smaller ECM islands than larger islands or conformal ECM surfaces. They then look at variants on the cells with talin/kindlin depleted. They showed that adhesion to the smaller ECM islands was not talin or kindlin dependant in contrast to standard understanding of adhesion to large ECM areas.

To this point, both cell types agreed well. Finally, the depleted pavilion and added back the talin head and showed the cell types responded differently - effecting specially enhanced adhesion of HeLa cells more so than fibroblasts.

From their data, they conclude that the specially enhanced adhesion state is fundamentally different from canonical adhesion and is not dependant upon mechanotransduction.

Overall, I find these to be interesting but bold claims without backup from imaging data. I would expect that super-resolution confocal microscopy could help to visualise the specially enhanced mechanism and to make sure that we are not looking at eg adhesion bridging events. Basic things like what do the microfilaments appear like as cells are forced into the specially enhanced state - do stress fibres not form and if mechanotransduction is not involved, does retrograde flow increase? It would also be interesting to try fibroblasts adhering to fibronectin vs vitronectin, where adhesion bridging is more regularly seen and where the cells can form larger adhesions in response to restricted ECM patterns (as shown in work by Duncan Sutherland and others).

I hope that the manuscript has attracted insightful reviews from more biophysical reviewers than myself - but, for me, I would like to see the paper including more classical cell biology for confirmation.

Version 2:

Reviewer comments:

Reviewer #1

(Remarks to the Author)

In the rebuttal letter and revised manuscript, the authors have thoroughly addressed all comments point-by-point.

With regards to my previous questions on the analysis: it was good to see a direct comparison of the original and proposed alternative analysis method, e.g. for single unbinding events, adhesion work etc. Since the shown differences do apparently not significantly influence the key findings and the limitations are also discussed in the manuscript now, there are not any further queries related to the analysis from my side.

Also, additional experiments were undertaken that strengthen the paper in my opinion. Importantly, the results of the myosin blocking experiments are interesting and support the original findings. I am confused, however, with the interpretation of the results in Fig. 6B (Fig. R1C). The authors state in the rebuttal letter "Hence, also the adhesion force per area to the largest

and smallest fibronectin pattern reduced, while it remained similar to the intermediate sized fibronectin pattern (Fig. R1D)", but (despite the shown p-values) the medians of blebbistatin-treated and controls for 2.4um² in R1D look rather similar to untreated controls? Please check the datasets again and also revise the figure legend of Fig. 6 to make it clear what the respective reference measurements in A and B are (I assume the legend should read untreated WT cells as in Fig.R1?).

Moreover, the imaging data provide interesting insights into the kinetics of adhesion formation in dependence of the adhesion pattern size. This, together with the more detailed analysis of the single unbinding events, improves interpretation of the findings.

Line 484 is confusing to me. If there are no clear signs of cluster formation within the analysed attachment period of the SCFS experiments, why do you conclude that clustering is one of the driving mechanisms? Although it might be true, this should be better discussed.

Generally, it remains still unclear to me whether one can confidently talk about an active sensing mechanism (e.g. "cells sense and respond", line 423), in particular when taking the relevance of the adhesion patterns circumference into account, which is nicely shown now. Nevertheless, the observation of a spatially enhanced adhesion is of interest, and might be even relevant for the discussed scenario of a native ECM environment.

While it is understandable that a comparison of talin levels in wt and TKO-THD fibroblasts is technically difficult, it would still be worth to present data confirming absence of talin in the talin KO fibroblasts using another antibody.

Typo line 290 (dependent).

Reviewer #2

(Remarks to the Author)

Reviewer #3

(Remarks to the Author)

Reviewer 1 and 2 combined have produced a range of highly appropriate biophysical comments to which the authors have responded to at length. I found their review very useful to myself.

I was not strongly supportive on first review, but I feel the authors have answered my questions as best they can and explained limitations of what they can achieve well.

If reviewers' 1 and 2 are happy with the response to their questions, I am satisfied.

Version 3:

Reviewer comments:

Reviewer #1

(Remarks to the Author)

All my points were addressed in the second revision, thank you.

Reviewer #2

(Remarks to the Author)

Reviewer #1 (Remarks to the Author)

Reviewer 1: In their manuscript, Wang et al. set out to study cell adhesion to spatially confined ECM ligands, with focus on early stages of cell adhesion formation as quantitated by single cell force spectroscopy. To spatially confine ECM ligands, they combine the AFM assay with micropatterned glass surfaces functionalised with collagen and fibronectin. Thereby patterns of increasing sizes are created, ranging from 2.4 μm^2 to 81 μm^2 , onto which single cells, immobilised on an AFM cantilever, are pushed for a defined contact time. As expected, they find that with increasing contact time and pattern size, overall cell adhesion forces increase. When they, however, normalise the overall adhesion forces to the respective contact areas, they find surprisingly over proportionally enlarged adhesion forces per area for the small patterns. They conclude from this that the studied cell types (HeLa cells and fibroblasts) can actively measure the size of the substrate area and switch to a “spatially enhanced adhesion state”. After conducting a set of experiments using integrin activating agents, talin and kindlin knockout cells, as well as cells overexpressing the talin head domain (supposedly increasing integrin activation), they come to the conclusion that cells enter this “spatially enhanced adhesion state” independently of talin or kindlin, while integrin activation, clustering and mechano-transduction pathways might be required, albeit in a cell-type dependent manner. In their manuscript, the authors present an extensive and overly interesting dataset that might point at a novel mechanism by which cells adjust their adhesion to spatially restricted ECM ligands. However, some of the main conclusions appear not fully supported by the presented data. Overly, the underlying mechanisms remain rather vaguely described. There are multiple questions and points that should be addressed for clarity.

Authors: Thank you for the positive and encouraging summary of our work. Below, we provide detailed responses to each point raised by the reviewer and outline the corresponding revisions made to the Manuscript. We believe that the additional experiments and analyses provided in this revised version strongly support our main conclusions.

Reviewer 1: The authors claim in the abstract that mechanistic insights were obtained (abstract line 31/32). They conclude “that the actin cytoskeleton along with integrin clustering and adhesome formation plays a cell and/or integrin type specific role in sensing ECM, responding, and state switching to adhere to the ECM protein area.”(line 374).

These statements, however, seem not well supported by the shown data. There were no experiments conducted where f-actin or actomyosin contractility was perturbed. For instance, experiments interfering with actomyosin contractility, as performed by

Changede and al ([https://www.cell.com/developmental-cell/fulltext/S1534-5807\(15\)00715-7#mmc1](https://www.cell.com/developmental-cell/fulltext/S1534-5807(15)00715-7#mmc1)) could be performed to support this statement.

Authors: Our results showing that the expression of the talin head domain in talin depleted HeLa cells or fibroblasts had different effects on the spatially enhanced adhesion led us to conclude “*that the actin cytoskeleton along with integrin clustering and adhesome formation plays a cell and/or integrin type specific role in sensing ECM, responding, and state switching to adhere to the ECM protein area.*”. The rationale for the conclusion on the role of the actin cytoskeleton, was that talin engages integrins to the actin cytoskeleton and takes essential roles in integrin clustering and mechanotransduction¹⁻⁵. The cell type and ECM protein dependent adhesion response in the absence of an integrin-cytoskeleton connection led us to this conclusion. However, we agree with the reviewer that such conclusion could be supported by additional experiments. Hence, we performed SCFS experiments of HeLa cells and fibroblasts in the presence of 20 μM blebbistatin to perturb myosin II contractility. We quantified the cell adhesion force to three different sizes of collagen I patterns ($\approx 81.2 \mu\text{m}^2$, $\approx 33.2 \mu\text{m}^2$, and $\approx 3.4 \mu\text{m}^2$, Fig. R1A,B) or fibronectin patterns ($\approx 58.1 \mu\text{m}^2$, $\approx 8.8 \mu\text{m}^2$, and $\approx 2.4 \mu\text{m}^2$, Fig. R1C,D).

Inhibition of myosin II lowered the adhesion force of HeLa cells to collagen I patterns having an area of $\approx 81.2 \mu\text{m}^2$ at contact times ≥ 120 s (Fig. R1A). However, the adhesion force of HeLa cells to smaller collagen I patterns remained unaffected by the inhibition of myosin II. Hence, the adhesion force per substrate area of HeLa cells to $\approx 81.2 \mu\text{m}^2$ large collagen I patterns slightly decreased at contact times ≥ 120 s (Fig. R1B). These results show that the canonical adhesion initiation of HeLa cells to collagen I depends on myosin II-mediated actin contractility at contact times ≥ 120 s, while the spatially enhanced adhesion of HeLa cells to collagen I is independent of actomyosin contractility within the first 360 s of contact.

The inhibition of myosin II reduced the adhesion force of fibroblasts to $\approx 58.1 \mu\text{m}^2$ fibronectin patterns for contact times ≥ 20 s and, surprisingly, also to the smallest ($\approx 2.4 \mu\text{m}^2$) fibronectin patterns for all contact times (Fig. R1C). In contrast, the adhesion force of fibroblasts to $\approx 8.8 \mu\text{m}^2$ fibronectin patterns remained unchanged. Hence, also the adhesion force per area to the largest and smallest fibronectin pattern reduced, while it remained similar to the intermediate sized fibronectin pattern (Fig. R1D). These results are in good agreement with those obtained using TKO+THD fibroblasts, which showed a similar behavior (see Fig. 6B, original Manuscript).

In addition to the experiments perturbing myosin II, we have perturbed F-actin using 1 μM latrunculin A, which diminished the adhesion force of both cell lines to all pattern sizes (Fig. R2A-D). Hence, an intact actin cytoskeleton is essential to initiate the canonical and the spatially enhanced cell adhesion.

Taken together, the additional experimental results strengthen our above mentioned conclusion and add that actomyosin contractility and hence mechanotransduction is cell line and/or integrin type specific in regulating the spatially enhanced cell adhesion. We have included and discussed the additional experimental data in the revised

Manuscript (see revised Fig. 6A,B, Supplementary Fig. 10A,B, Supplementary Fig. 12, and Supplementary Fig. 13A,B, Methods section “Single-cell force spectroscopy (SCFS)”, and Results section “Cell-specific integrin-actin engagement and integrin-mediated ECM sensing”).

Furthermore, to avoid confusion of the reader, we have revised the sentence cited by the reviewer in our Manuscript, and now clarify that our experiments show that both cell lines require an intact actin cytoskeleton to conduct spatially enhanced adhesion. The revised sentence reads “The results, thus, indicate that the actomyosin contractility, integrin clustering and adhesome formation may play a cell and/or integrin type specific role in sensing, responding, and states switching to adhere to the ECM protein area” (see revised Discussion).

Figure R1. The spatially enhanced adhesion state is independent of myosin II contractility. (A-D) Adhesion force and adhesion force per area of 20 μM blebbistatin-treated (A,B) wt HeLa cells and (C,D) wt fibroblasts to different areas of collagen I or fibronectin patterns at given contact times. Dots represent adhesion forces of individual cells, red bars the median, and n (cells) the number of individual cells tested in at least three independent experiments. (A,C) Adhesion force and (B,D) adhesion force per area of untreated wt HeLa cells or wt fibroblasts are given as reference (semitransparent). P values were calculated by two sided Mann-Whitney tests and compare the given adhesion forces with the reference data.

Figure R2. The spatially enhanced adhesion state requires actin cytoskeleton. (A-D) Adhesion force and adhesion force per area of $1 \mu\text{M}$ latrunculin A-treated (A,B) wt HeLa cells and (C,D) wt fibroblasts to different areas of collagen I or fibronectin patterns at given contact times. Dots represent adhesion forces of individual cells, red bars the median, and $n(\text{cells})$ the number of individual cells tested in at least three independent experiments. (A,C) Adhesion force and (B,D) adhesion force per area of untreated wt HeLa cells or wt fibroblasts are given as reference (semitransparent). P values were calculated by two sided Mann-Whitney tests and compare the given adhesion forces with the reference data.

Reviewer 1: If integrin clustering was involved, this should also be visible from analysing the single rupture events (especially at prolonged contact times). However, the presented data are not explained well enough (e.g. for which contact time were these rupture forces analysed in Supplementary figure 5?) to understand this. In addition, analysis of the single unbinding events at prolonged contact times might help to understand whether clusters had formed, since this should result in an increase of the smallest rupture events. It would be interesting to compare whether an eventual increase of individual rupture events already occurs at earlier time points on smaller areas.

Authors: Thank you for pointing out the lack of clarity in the description of the data in the figure legends. For the analysis of rupture and tether force shown in Supplementary Fig. 5 (original Manuscript) we pooled the force-distance curves across all contact times for each pattern size. We have revised the legend of the Figure to describe the conditions at

which the data were recorded more clearly. We have included in the Methods section “*Analyzing single rupture and tether forces from SCFS data*” in our revised Manuscript a description that we pooled all contact times for the rupture analysis since we have previously demonstrated that for fibroblasts adhering to fibronectin the rupture forces are independent of the contact time⁶. To verify that rupture forces are indeed contact time independent, also in our experimental conditions, we analyzed the rupture force of HeLa cells during the detachment process from collagen I patterns (Fig. R3A,B) and of fibroblasts detaching from fibronectin patterns (Fig. R3C,D) for each contact time. The analysis of the median rupture force shows a contact time dependency, with an increase of the median rupture force at contact times > 20 s for HeLa cells detaching from collagen I and > 60 s for fibroblasts detaching from fibronectin (Fig. R3A,C). This increase could be due to 1) changes in the integrin-ligand bond characteristics due to integrin clustering, or 2) increased numbers of unresolved simultaneously unbinding of multiple integrins. Hence, we analyzed the rupture force distributions of HeLa cells detaching from collagen I patterns at each experimental contact time (Fig. R3B). We find that the maxima of fitted Gaussian distributions range between ≈ 33 and ≈ 36 pN, with no significant differences between the maxima. Furthermore, we could not find a correlation between contact time and the maxima of the Gaussian fits. The rupture force distributions of fibroblasts detaching from fibronectin patterns showed multiple peaks, which we fitted using double Gaussian functions (Fig. R3D). The first maxima of the Gaussian fits are in a range of ≈ 36.5 and 38 pN, which show no significant difference. Further, the second peaks appear at approximately twice this force range, indicating that these are two integrins unbinding at the same time. From these results, we interpret that the unbinding force of single integrins is similar among the contact times. Further, the increase of the median rupture force (Fig. R3C) results from a stronger contribution of the temporally unresolved unbinding of multiple integrins from fibronectin.

Importantly, especially for fibroblasts, we observe the simultaneous unbinding of multiple integrins to become more frequent with increasing contact time. While this may reflect on integrin clustering, we cannot spatially or temporarily resolve the unbinding events of individual integrins in our experimental setup. Hence, the simultaneous unbinding of integrins may occur at very different positions of the contact area between the cell and the ECM protein pattern. Thus, we prefer not to interpret these results as increased clustering, since this would be, in our opinion, rather speculative.

Figure R3. Rupture forces of HeLa cells detaching from collagen I patterns and of fibroblasts detaching from fibronectin patterns are contact time independent. (A-D) Rupture force of (A,B) HeLa cells detaching from collagen I patterns or of (C,D) fibroblasts detaching from fibronectin patterns. For rupture force analysis, the force-distance curves recorded were pooled across all pattern sizes for each contact time indicated. (A,C) Single dots represent individual rupture forces, red bars their median. P values comparing given rupture forces with rupture forces at 5 s contact time were calculated using two sided Mann-Whitney tests. (B,D) Histograms with a bin size of 5 pN show the distribution of rupture forces. Lines show fits of (B) single or (D) double Gaussian functions to rupture force distributions from (B) 25 pN to 50 pN or (D) 25 pN to 90 pN. P values comparing the given (B) mean values derived from single Gaussian functions or (D) the first means of double Gaussian functions and were calculated using extra sum-of-squares F -tests.

Reviewer 1: In addition, the formation of larger adhesion sites (e.g. fluorescently tagged paxillin or vinculin recruitment) could be supported by imaging data, e.g. combining AFM and confocal or TIRF microscopy. Alternatively, they could perform live imaging after adding the cells over the micropatterned surface and imaging cells as soon as they start to attach.

Authors: The reviewer asks for imaging data to visualize the formation of larger adhesion sites. To address how larger adhesion sites assemble, we attached single, rounded paxillin-GFP expressing HeLa cells (Fig. R4A) or fibroblasts (Fig. R4B) to a ConA-coated AFM cantilever and brought the cells into contact with collagen I or fibronectin patterns of different sizes (areas). During contact times of up to ≈ 70 min, we monitored paxillin every 2.5 min by confocal microscopy. In HeLa cells that are in contact with $\approx 81.2 \mu\text{m}^2$ collagen I patterns for ≈ 7.5 min contact time, paxillin-GFP starts to localize at the periphery of the collagen I pattern. With increasing contact time, the intensity of paxillin-GFP increased both at the periphery and in the center of the pattern. Apparently, more paxillin-GFP localized at the periphery of the collagen I pattern. On the $\approx 33.2 \mu\text{m}^2$ collagen I pattern, we observed paxillin-GFP to localize in HeLa cells at the periphery of the pattern, but with slower dynamics. After ≈ 15 min of contact time, HeLa cells

adhering to the $\approx 33.2 \mu\text{m}^2$ collagen I pattern, started to assemble paxillin-GFP only in some peripheral regions. However, the paxillin-GFP covered the entire periphery after ≈ 37.5 min of contact time. No clear increase of fluorescent intensity of paxillin-GFP was observed in the center of the pattern. Interestingly, on the smallest collagen I pattern, we observed paxillin-GFP at some peripheral regions after ≈ 7.5 min, while after ≈ 15 min contact time paxillin-GFP covered the entire periphery of the collagen I pattern.

In fibroblasts, paxillin-GFP localized a bit more diffuse compared to HeLa cells, which is likely due to the low amount of paxillin-GFP on top of the endogenous paxillin expression^{7,8}. However, in fibroblasts paxillin also localized at the periphery of the fibronectin patterns. On the largest fibronectin pattern ($\approx 58.1 \mu\text{m}^2$) paxillin started localizing at the periphery at contact times ≈ 7.5 min. Similar to HeLa cells on collagen I patterns, the localization of paxillin to the intermediate sized fibronectin pattern of $\approx 8.8 \mu\text{m}^2$ was observed from ≈ 22.5 min contact time and hence delayed compared to the largest fibronectin pattern. On the smallest fibronectin pattern of $\approx 2.4 \mu\text{m}^2$, the fibroblasts started recruiting paxillin already at ≈ 7.5 min contact time.

In summary, we observe paxillin recruitment mainly to the periphery of the ECM protein patterns for HeLa cells adhering to collagen I patterns and for fibroblasts adhering to fibronectin patterns. From our timelapse microscopy experiments we conclude that the dynamics of the paxillin recruitment depends on the pattern size. Notably, paxillin recruitment is observed only after much longer contact times than those probed in SCFS experiments (see Fig. 6C,D, original Manuscript). Hence, projecting the results of SCFS to those revealed by confocal microscopy would be rather speculative. Furthermore, the localization of paxillin does not clearly indicate how it influences adhesion, whether it enhances or reduces adhesion forces. Nevertheless, to illustrate how long-term adhesion, *i.e.*, paxillin recruitment, of HeLa cells and fibroblasts is influenced by the ECM protein pattern size, we included the data shown in Fig. R6 in Supplementary Fig. 8 and discussed the data in the Results section “*Long term adhesion dynamics depend on the ECM protein pattern area*” in the revised Manuscript.

Figure R4. The formation of adhesion sites by HeLa cells on collagen I patterns and by fibroblasts on fibronectin patterns depends on the pattern size. (A,B) Timelapse confocal microscopy images of paxillin-GFP expressing (A) HeLa cells on collagen I patterns and (B) of fibroblasts on fibronectin patterns. Pattern size (area) and contact time are indicated. $n = 5$ independent experiments for each condition. A single, rounded paxillin-GFP expressing (A) HeLa cell or (B) fibroblast was attached to a ConA-coated AFM cantilever and then brought into contact with collagen I or fibronectin patterns until reaching contact times of ≈ 70 min. Collagen I or fibronectin patterns were labelled using Alexa fluor 555 (gray). Scale bars, 5 μm .

Reviewer 1: As it appears, the number of bound integrins is estimated from the single rupture forces determined (dataset in Supplementary Fig 5) and the overall adhesion force. This appears over-simplified, since the steps as shown in the force curve example (Supplementary Figure 5) might not be simply add up to the maximal force due to spring-like force loadings in between. In addition, the tether forces would contribute significantly to the overall detachment forces. Since the authors also analyse the single rupture and tether events, it appears more appropriate to present the actual number of rupture events per curve (then also rupture and tethers could be distinguished).

Authors: We agree with the reviewer that our approach is simplified, which is why we state in the Manuscript that we estimate the number of ligand-bound integrins. The reviewer is correct that estimating the rupture events from the overall cell adhesion force presents a rough estimate. However, due to the spring-like loading of cantilever and cell during the mechanical detachment process⁹⁻¹³, this approach underestimates the number of rupture events and thus of ligand-bound integrins. We deliberately consider tether events for estimating the number of ligand-bound integrins since single membrane tethers are most likely linked to the ECM protein by at least one integrin¹³⁻¹⁵. However, since tether forces are higher than the rupture forces, our rough approach overestimates the number of ligand-bound integrins in the part of the force-distance curve that is dominated by tether events (under the assumption that one integrin links the tether to the substrate).

Unfortunately, the suggestion of the reviewer to count unbinding events and tethers per force-distance curve is not feasible. As exemplified in Supplementary Fig. 5A (original Manuscript), force-distance curves often show rupture events with rupture forces >150 pN. Such higher rupture forces clearly originate from the simultaneous rupture of multiple ligand-integrin bonds. However, in these large events the individual rupture events of single integrins from their ligands remain temporally unresolved ligand (see also Fig. R3)^{6,16,17}. The wide distribution of single integrin rupture forces (Fig. R3) makes it impossible to precisely determine how many integrins unbound from the ligand at rupture forces >150 pN. Hence, the approach to count unbinding events will also not provide an accurate number of ligand-bound integrins.

However, to compare our approach with the one suggested by the reviewer, we analyzed force-distance curves of HeLa cells adhering to collagen I or of fibroblasts adhering to fibronectin substrates and counted the number of single rupture events as described above. Alternatively, we used the same experimental data set to estimate the number of ligand-bound integrins by dividing the cell adhesion force through the mean rupture force of single integrins (Fig. R5). The comparison of both approaches shows that counting the

number of single rupture events considerably lowered the number of ligand-bound integrins per substrate area compared to our approach to estimate the number of ligand bound integrins. This difference between both approaches increases with contact time with which the cell adhesion force increases and the thus the number of large rupture events that mask the rupture events of individual integrins from the ligand (Fig. R3). Hence, we decided to keep our approach to estimate the number of integrin-ligand bonds in the original Manuscript. However, we have revised the Manuscript to clearly describe the limitations of our approach in estimating the number of integrin-ligand bonds (see Results section “*With reducing ECM protein area cells switch to a spatially enhanced adhesion state*”).

Figure R5. The density of ligand-bound integrins is underestimated by counting rupture events recorded in single force-distance curves compared to dividing the cell adhesion force by the average rupture force. (A,B) Ligand-bound integrins per substrate area for (A) wt HeLa cells adhering to collagen I and (B) wt fibroblasts adhering to fibronectin patterns. Left, ligand-bound integrin densities as estimated by counting rupture events per patterned substrate area in detachment force-distance curves. Right, ligand-bound integrin densities as estimated by dividing the cell adhesion force per patterned substrate area (data taken from Fig. 4, original Manuscript) by the average rupture force of single integrins. Rupture forces from HeLa cell detaching form collagen I or from fibroblasts detaching from fibronectin substrates were taken from Supplementary Fig. 5C,D, original Manuscript. Dots represent densities of ligand-bound integrins of single cells, red bars median values, and $n(\text{cells})$ the number of individual cells tested in at least three independent experiments. P values were calculated using two-sided Mann-Whitney tests and compare the ligand-bound integrin density estimated by counting (left) per area and dividing adhesion force per area by average rupture force (right).

Reviewer 1: Given the not insignificant contribution of unspecific adhesion, it appears logic to correct for the non-specific reference (as done in Suppl fig. 4). From the methods it appears that a constant median value on passivated surfaces was simply subtracted. Would it not be more accurate to subtract proportional to the area of unspecific coating that the cell contacts on each of the patterns (which becomes relatively larger for the smaller patterns)? There could potentially arise a bias for larger patterns that were mostly in contact with the ECM ligand but not the surrounding passivated surface. Is this effect taken into account by the explained method?

Authors: The reviewer is correct, we corrected for the specific cell adhesion force per ECM protein area by subtracting the median unspecific adhesion force to either BSA or FNIII7-10ΔRGD from the measured adhesion force. We did not consider the ratio of ECM protein and passivation in the contact area in our correction. This has mainly two reasons. First, BSA added for passivation can also partially co-localize at the collagen I pattern (see Fig. 1E, original Manuscript) and fibroblasts can also establish unspecific adhesion to fibronectin^{16,18}, which is closely resembled by the cell adhesion to FNIII7-10ΔRGD. Second, the actual ratio between specific and unspecific ligands largely depends on the contact area between the cell and the substrate. Since the contact area varies between $\approx 85 \mu\text{m}^2$ and $\approx 135 \mu\text{m}^2$ for HeLa cells and $\approx 70 \mu\text{m}^2$ and $\approx 130 \mu\text{m}^2$ for fibroblasts we cannot precisely determine the cell specific ratio between passivated and ECM protein coated area. We thus decided to follow a conservative approach in which we subtract the maximum unspecific adhesion force from the cell adhesion force. We are aware that this approach provides a lower specific adhesion force, which increases with the ECM pattern area. However, the effect of subtracting the maximum unspecific adhesion force per area has only a minimal effect on the adhesion force per area on the large pattern.

Since we understand the concern of the reviewer that this approach may have biased the results towards lower adhesion forces per area on large ECM protein patterns, we refined the correction as suggested by the reviewer. In this correction, we used the mean contact area of HeLa cells and fibroblasts to the substrate and evaluated the ratio of ECM protein and passivated surface. We used this ratio to subtract a substrate area dependent unspecific adhesion force from the adhesion force to the patterned surfaces. As depicted in Fig. R6A,B, this ratiometric correction only marginally changes the adhesion force of HeLa cells or fibroblasts. Importantly, the main observations made in our Manuscript remain valid: If the contact area to collagen I is $\geq 42.6 \mu\text{m}^2$, HeLa cells establish maximum adhesion force to collagen I, while they establish smaller adhesion force if the contact area to collagen I is smaller. Similarly, fibroblasts in contact to fibronectin patterns $\geq 34.1 \mu\text{m}^2$ establish maximum adhesion force, while they establish lower adhesion force if the area of the fibronectin pattern is smaller. On the other hand, HeLa cells establish similar adhesion force per area to collagen I patterns having areas of $\approx 81.2 \mu\text{m}^2$, $\approx 42.6 \mu\text{m}^2$ and $\approx 33.2 \mu\text{m}^2$, but increase the adhesion force per collagen I area 7-fold to $\approx 4.3 \mu\text{m}^2$ large collagen I patterns and 6-fold to $\approx 3.4 \mu\text{m}^2$ large collagen I patterns (Fig. R6C). Fibroblasts establish similar adhesion force per fibronectin area to $\approx 58.1 \mu\text{m}^2$ and $\approx 34.1 \mu\text{m}^2$ large fibronectin patterns and increase the adhesion force per area to ≈ 2 -fold on $\approx 8.8 \mu\text{m}^2$, 6-fold to $\approx 5.3 \mu\text{m}^2$, and 8-fold to $\approx 2.4 \mu\text{m}^2$ large fibronectin patterns (Fig. R6D). Hence, we conclude that our initial correction and the suggested correction by the reviewer do not change the results and thus the outcome of our Manuscript. Due to the reasons mentioned above, we decided to keep the correction for the unspecific cell adhesion as described in the original Manuscript. Thank you, your comment guided us to test correction of the unspecific adhesion force by an alternative model.

Figure R6. Correcting the cell adhesion force for unspecific adhesion contributions proportional to the area of the unspecific coating does not change the results. (A,B) Corrected adhesion force of (A) wt HeLa cells and (B) wt fibroblasts to collagen I or fibronectin patterns having different areas and measured at different contact times. The measured adhesion force (data taken from Fig. 3, original Manuscript) were corrected for contribution of unspecific adhesion force to BSA or FNIII7-10ΔRGD proportionally to the mean contact area (data taken from Fig. 2D, Supplementary Fig. 1E, original Manuscript). Dots represent adhesion forces of single cells, red bars median values, and $n(\text{cells})$ the number of independent cells tested in at least three independent experiments. Adhesion force of (A) wt HeLa cells or (B) wt fibroblasts to indicated protein patterns before correction is given in the background as reference (semitransparent). (C,D) Adhesion force per area of (C) wt HeLa cells and (D) wt fibroblasts (data taken from Fig. 4A,B, original Manuscript) normalized to the unit area of the ECM protein pattern (data taken from Fig. 2D, original Manuscript) at given contact times. Dots represent adhesion forces of single cells per unit area of ECM protein pattern, red bars median values, and $n(\text{cells})$ the number of independent cells tested in at least three independent experiments. Adhesion force per unit ECM area of (A) wt HeLa cells or (B) wt fibroblasts to indicated protein patterns before correction is given in the background as reference (semitransparent). P values were calculated by two sided Mann-Whitney tests and compare the displayed and reference cell adhesion force.

Reviewer 1: Are there any geometric considerations that should be taken into account? E.g., the process of detaching/“peeling off” the cell from the substrate could be different for larger and smaller areas. Since the cell contact area probably “shrinks” from the outside to the inside during the detachment process, would the results be different if the adhesion energy and not the overall detachment force would be plotted? In any case, it would be relevant to plot the adhesion work/energy over time.

Authors: The reviewer asks whether any geometric considerations should be taken into account. In our experiments, we have carefully placed the cantilever bound cells above the patterned surfaces to ensure that the cells are in contact with the entire ECM protein pattern (see Supplementary Fig. 1D, original Manuscript). Since the contact area of the cell is larger than the ECM protein pattern, the contact area geometry with the pattern is similar, with the cell being in flat contact with the ECM protein pattern. However, the contact geometry between the unrestricted ECM proteins and the largest pattern may be different due to the curved cell edge being in contact with ECM proteins. Because in the latter conditions the adhesion forces measured to the unrestricted substrates and to the larger patterned substrates are similar, we can conclude that the contact geometry does not affect the switch to the spatially enhanced adhesion state.

The reviewer further asks whether the results change if we quantify the adhesion work/energy, which is the integral of the adhesion force over the distance, instead of the adhesion force. We agree that the adhesion work/energy would be an interesting parameter to describe the de-adhesion process of cells from substrates. However, the work/energy of cell adhesion is substantially influenced by a variety of cell properties, including elasticity, deformation, or dissipation. Upon detachment from the substrate, the rather soft cell is considerably stretched and deformed over the distance of several tens of micrometers^{10–12,15}. Thus, the ‘cell adhesion work/energy’, represents the convolution of the mechanical deformation of the cell and the mechanical stress applied to the integrin-mediated adhesive bonds formed with the substrate. Consequently, the ‘cell adhesion work/energy’ required to detach a cell from a support largely depends on other parameters than cell adhesion and thus can be difficult to interpret^{12,15,19}. In fact, a

cell that establishes high adhesion force, but is very little deformed (stiff) during the detachment process from the substrate, can show a smaller ‘cell adhesion work/energy’ than a cell that adheres to the support with lower force, but is strongly deformed (soft) during the detachment process (Fig. R7). This difference of the work/energy can even shift further, if the cell during the detachment process produces many several micrometer long membrane tethers being pulled out from the cell membrane²⁰.

Figure R7. Adhesion work/energy is a convolute measure of adhesion force and mechanical deformation of the cell during the mechanical detachment of the soft cell from the substrate. Depicted are two hypothetical retraction force-distance curves in SCFS experiments. The blue curve displays a cell with high adhesion force that is very little deformed during the mechanical detachment process. The purple curve represents the mechanical detachment process of a cell establishing less adhesion force but being strongly deformed during the separation of cell and substrate (as shown in Fig. R8A, HeLa cells establish considerably higher adhesion work/energy to BSA than to $3.4 \mu\text{m}^2$ large collagen I patterns). While the blue force-distance curve shows a nearly 2.5-fold higher adhesion force than the purple force-distance curve, the adhesion energy of the blue force-distance curve is only about half of the purple. This ratio increases even further if during the detachment process numerous membrane tethers are extracted from the cell

membrane. Hence, the adhesion energy is not a suitable metrics to quantify the adhesion of cells to an ECM substrate.

Nevertheless, intrigued by the reviewer’s comment we quantified the adhesion work/energy for HeLa cells adhering to collagen I patterns and for fibroblasts adhering to fibronectin patterns (Fig. R8A,B). HeLa cells established similar adhesion work/energy to unrestricted collagen I substrates and to printed collagen I patterns having areas of $\approx 81.2 \mu\text{m}^2$, $\approx 42.6 \mu\text{m}^2$, and $\approx 33.2 \mu\text{m}^2$, and considerably reduced the adhesion work/energy to collagen I patterns having areas $< 33.2 \mu\text{m}^2$ (Fig. R8A). Thereby, the adhesion work/energy of HeLa cells was similarly high to $\approx 81.2 \mu\text{m}^2$ and $\approx 42.6 \mu\text{m}^2$ large collagen I patterns and similarly low to collagen I patterns having areas of $\approx 4.3 \mu\text{m}^2$ and $\approx 3.4 \mu\text{m}^2$. The adhesion work/energy to $\approx 33.2 \mu\text{m}^2$ collagen I patterns was between the high and the low adhesion work/energies (Table R1). The adhesion work/energy of HeLa cells to BSA was similar to the adhesion work/energy to the largest collagen I pattern, despite drastic differences in adhesion force (see Fig. 3A, original Manuscript). This difference between adhesion work/energy and force is due to the very different attachment mechanisms of HeLa cells to BSA compared to collagen I (*i.e.*, unspecific *versus* specific), which leads to very different detachment processes, such as exemplified above (Fig. R7). Nevertheless, the adhesion work/energy per substrate area followed a very closely the trend described by the adhesion force per area. While it was very low on the collagen I patterns having areas of $\approx 81.2 \mu\text{m}^2$, $\approx 42.6 \mu\text{m}^2$ and $\approx 33.2 \mu\text{m}^2$, it was 4- to 10-fold higher on collagen I patterns having areas of $\approx 4.3 \mu\text{m}^2$ and $\approx 3.4 \mu\text{m}^2$ (Fig. R8C).

Fibroblasts established similar adhesion work/energy to unrestricted fibronectin substrates and $\approx 58.1 \mu\text{m}^2$ large fibronectin patterns. Fibroblasts initiating adhesion to

34.1 μm^2 large fibronectin patterns had a tendency towards lower adhesion work/energy compared to unrestricted fibronectin substrates, which was statistically significant at 5 s and 360 s (Fig. R8B and Table R2). The adhesion work/energy of fibroblasts initiating adhesion to the fibronectin patterns of $\approx 8.8 \mu\text{m}^2$, $\approx 5.3 \mu\text{m}^2$ or $\approx 2.4 \mu\text{m}^2$, were lower compared to unrestricted fibronectin substrates (Table R2). Interestingly, the adhesion work/energy of fibroblasts was independent of the fibronectin pattern area at contact times ≤ 60 s. At longer contact times the adhesion work/energy was higher on $\approx 58.1 \mu\text{m}^2$ large fibronectin patterns compared to $\approx 8.8 \mu\text{m}^2$, $\approx 5.3 \mu\text{m}^2$ or $\approx 2.4 \mu\text{m}^2$ large patterns. Fibroblasts showed similar adhesion work/energy on the small fibronectin patterns ($\approx 8.8 \mu\text{m}^2$, $\approx 5.3 \mu\text{m}^2$, or $\approx 2.4 \mu\text{m}^2$). Importantly, the adhesion work/energy per area of fibroblasts initiating adhesion to fibronectin patterns of the different sizes inversely correlated to the pattern area (Fig. R8D). Similar as the adhesion forces per area, the adhesion work/energy per area is 10- to 20-fold higher on the $\approx 2.4 \mu\text{m}^2$ large fibronectin pattern compared to the $\approx 58.1 \mu\text{m}^2$ large fibronectin pattern.

In summary, the adhesion work/energy per substrate area of HeLa cells adhering to collagen I patterns and of fibroblasts adhering to fibronectin patterns closely follows the adhesion force per area described in the original Manuscript. Since the measured adhesion work/energy is not a direct measure of adhesion of the cell but also involves considerable mechanical deformation of the soft cell (Fig. R7), we prefer to display adhesion force as a direct measure of cell adhesion, which is easier to interpret^{10,21}.

Figure R8 (previous page). Adhesion work/energy of HeLa cells and fibroblasts correlates with the area of the patterned ECM protein substrate. (A-D) Adhesion work/energy and adhesion work/energy per area of (A,C) wt HeLa cells to collagen I or (B,D) wt fibroblasts to fibronectin patterns having different areas and measured at different contact times. After printing the ECM protein patterns, the remaining uncoated glass surfaces were passivated with (A,C) BSA or (B,D) FNIII7-10ΔRGD to minimize unspecific cell adhesion to glass. Dots represent adhesion work/energy or adhesion work/energy per area of single cells, red bars median values, and $n(\text{cells})$ the number of independent cells tested in at least three independent experiments. (A,B) Adhesion work/energy of (A) wt HeLa cells to unrestricted collagen I or (B) wt fibroblasts to unrestricted fibronectin is given as reference (semitransparent). (C,D) Adhesion work/energy per area of (C) wt HeLa cells to $\approx 81.2 \mu\text{m}^2$ large collagen I patterns or (D) wt fibroblasts to $\approx 58.1 \mu\text{m}^2$ large fibronectin patterns is given as reference (semitransparent). P values were calculated using two sided Mann-Whitney tests and compare cell adhesion work/energy or adhesion work/energy per area of displayed data (black) and reference data. Statistical analysis of potential difference in adhesion work/energy established by wt HeLa cells to collagen I patterns and wt fibroblasts to fibronectin patterns of different sizes is given in Table R1 and R2.

Comparison of the adhesion work/energy of wt HeLa cells to printed collagen I patterns							
Patterned area (μm^2)		81.2					
Contact time (s)		5	20	60	120	240	360
Patterned area $42.6 \mu\text{m}^2$	P value	0.626	0.641	0.573	0.718	0.728	0.347
Patterned area $33.2 \mu\text{m}^2$	P value	0.579	0.991	0.789	0.170	0.641	0.284
Patterned area $4.3 \mu\text{m}^2$	P value	0.055	<0.001	0.047	0.002	0.014	<0.001
Patterned area $3.4 \mu\text{m}^2$	P value	0.004	<0.001	0.036	0.002	0.001	<0.001
Patterned area (μm^2)		42.6					
Contact time (s)		5	20	60	120	240	360
Patterned area $33.2 \mu\text{m}^2$	P value	0.988	0.497	0.459	0.356	0.421	0.900
Patterned area $4.3 \mu\text{m}^2$	P value	0.123	0.001	0.002	0.001	0.003	0.035
Patterned area $3.4 \mu\text{m}^2$	P value	0.025	<0.001	0.004	0.001	<0.001	0.001
Patterned area (μm^2)		33.2					
Contact time (s)		5	20	60	120	240	360
Patterned area $4.3 \mu\text{m}^2$	P value	0.179	<0.001	0.061	0.018	0.090	0.043
Patterned area $3.4 \mu\text{m}^2$	P value	0.085	<0.001	0.088	0.020	0.014	0.001
Patterned area (μm^2)		4.3					
Contact time (s)		5	20	60	120	240	360
Patterned area $3.4 \mu\text{m}^2$	P value	0.279	0.630	0.990	0.968	0.740	0.061

Table R1. Statistical analysis comparing the adhesion work/energy of wt HeLa cells to printed collagen I patterns with indicated substrate area at given contact time. P values compare adhesion work/energy of wt HeLa cells to printed collagen patterns of given area with that measured of the indicated collagen I pattern area at the given contact time. P values were calculated using two-sided Mann-Whitney tests, black values show significant difference ($P < 0.05$), and red values non-significant difference ($P \geq 0.05$). Data taken from Fig. R8A.

Comparison of the adhesion work/energy of wt fibroblasts to printed fibronectin patterns							
Patterned area (μm^2)		58.1					
Contact time (s)		5	20	60	120	240	360
Patterned area $34.1 \mu\text{m}^2$	P value	0.396	0.928	0.935	0.993	0.063	0.104
Patterned area $8.8 \mu\text{m}^2$	P value	0.685	0.213	0.013	0.098	0.001	0.003

Patterned area 5.3 μm^2	P value	0.652	0.773	0.539	0.097	<0.001	0.002
Patterned area 2.4 μm^2	P value	0.958	0.414	0.212	0.148	<0.001	<0.001
Patterned area (μm^2)		34.1					
Contact time (s)		5	20	60	120	240	360
Patterned area 8.8 μm^2	P value	0.725	0.142	0.607	0.048	0.214	0.395
Patterned area 5.3 μm^2	P value	0.200	0.718	0.217	0.036	0.148	0.287
Patterned area 2.4 μm^2	P value	0.568	0.310	0.748	0.073	0.049	0.019
Patterned area (μm^2)		8.8					
Contact time (s)		5	20	60	120	240	360
Patterned area 5.3 μm^2	P value	0.231	0.213	0.055	0.665	0.994	0.863
Patterned area 2.4 μm^2	P value	0.691	0.594	0.322	0.597	0.345	0.061
Patterned area (μm^2)		5.3					
Contact time (s)		5	20	60	120	240	360
Patterned area 2.4 μm^2	P value	0.440	0.414	0.492	0.833	0.233	0.098

Table R2. Statistical analysis comparing the adhesion work/energy of wt fibroblasts to printed fibronectin patterns with indicated substrate area at given contact time. *P* values compare the adhesion work/energy of wt fibroblasts to printed fibronectin pattern of given area with that measured of the indicated fibronectin pattern area at the given contact time. *P* values were calculated using two-sided Mann-Whitney tests, black values show significant difference ($P < 0.05$), and red values non-significant difference ($P \geq 0.05$). Data taken from Fig. R8B.

Reviewer 1: Was the overall surface contact area also quantitatively measured for the different patterns, e.g. using a combined confocal AFM setup? It might be possible that the cell shows distinct spreading on the different substrates at longer contact times. This might affect overall adhesion forces, since the cells also adhere to some extent with the surrounding area (e.g. “unspecific adhesion to BSA”). From Supplementary figure 1E it is not clear whether the graph presents different patterns that are pooled?

Authors: The reviewer asks whether the contact area was measured for the different ECM patterns. We have quantified the contact area of the cells on ECM protein patterns having different sizes (Supplementary Fig. 1D,E, original Manuscript). Cells were imaged directly after being in contact with the substrate.

The reviewer further comments that “*It might be possible that the cell shows distinct spreading on the different substrates at longer contact times*”. We kindly remind that in our SCFS measurements we characterized the adhesion force of cells being for in contact with the substrate for up to 360 s (contact time). During this relatively short time the cells remained in the rounded state and initiated adhesion. Accordingly, we have not observed cell spreading to the tested substrates. To validate this observation, we fluorescently labeled single rounded HeLa cells and fibroblasts using CellTracker, attached them to an AFM cantilever, and brought them in contact with unrestricted collagen I, BSA, fibronectin, or FNIII7-10 Δ RGD substrates until reaching a contact force of 2 nN, such as applied in our SCFS experiments (Fig. R9A). We then quantified the contact area between the cantilever bound cell and the substrate over the contact time.

The results show that during the contact time of 360 s the contact area slightly increases $\approx 7\%$ for HeLa cells and $\approx 9\%$ for fibroblasts. However, besides such minimal increase in contact area the cells did not show any spreading. Importantly, the contact area does not depend on the substrate (*i.e.*, collagen I or BSA for HeLa cells and fibronectin or FNIII7-10 Δ RGD for fibroblasts; Fig. R9B,C).

Since we cannot observe differences in the contact area on unrestricted ECM proteins and passivating proteins (BSA or FNIII7-10 Δ RGD), we conclude that the contact area on ECM protein patterns remains similar. Additionally, it is important to note that in SCFS experiments the contact area between cells and substrate is always larger than the pattern sizes/area (see Supplementary Fig. 1D, original Manuscript; also reviewer comment addressed earlier). Hence, the minimal changes in contact area would not change the specific interaction between cells and ECM protein patterns, rather it could alter the unspecific interactions to the passivating surface surrounding the ECM protein patterns. Importantly, we can exclude that potential differences in cell-substrate contact area on the patterned substrates cause the differences in adhesion forces per area we measured on the substrates. We have included these data in Supplementary Fig. 1E,F of our revised Manuscript.

Figure R9. Contact area of HeLa cells to collagen I or BSA and of fibroblasts to fibronectin or FNIII7-10 Δ RGD slightly increases with contact time. (A) Representative timelapse confocal images of CellTracker labeled single rounded HeLa cell or fibroblast attached to a ConA-coated cantilever in contact with the given substrate for the indicated contact time ($n_{\text{cells}}=10$). Average change and SD of contact area are indicated in percent for each contact time characterized. *P* values were calculated by two sided paired *t*-tests and compare the contact area at the given contact time with the contact area at 0 s contact time. Scale bar, 10 μm . (B,C) Contact area of HeLa cells and fibroblasts to indicated substrate at 0 s contact time. Dots represent contact areas of single cells, red bars median values, and $n(\text{cells})$ the number of individual cells measured. *P* values were calculated by two sided Mann-Whitney tests and compare the contact area of HeLa cells to collagen I and BSA or of fibroblasts to fibronectin and FNIII7-10 Δ RGD.

Reviewer 1: How were the paxillin and talin knockout cells created? And, were they thoroughly characterised with regards to integrin expression? If they were expanded from single cell clones, there might be intrinsic differences in integrin expression and/or

surface levels. This should be checked, since many conclusions rely on a direct comparison of WT and KO cell lines.

Authors: The talin KO fibroblasts were created and characterized in the Fässler lab and described previously²². Talin KO HeLa cells were created and characterized in the Müller lab and described previously²³. In these publications, the integrin expression levels were described to be similar to wild-type fibroblasts. Paxillin KO HeLa cells and paxillin KO fibroblasts were created using CRISPR/Cas9 as described in the Methods section “*Cell line engineering*” We have revised the Methods section to describe that paxillin KO HeLa cells and paxillin KO fibroblasts were expanded from single cell clones. We have now evaluated the expression levels of relevant integrins (integrin subunits $\alpha 1$, $\alpha 2$, and $\beta 1$ for HeLa cells and integrin subunits $\alpha 5$, αV and $\beta 1$ for fibroblasts) of wild-type and engineered cell lines using flow cytometry (Fig. R10A,B). Flow cytometry shows that the used HeLa cell lines express similar levels of the integrin $\alpha 1$ subunit on their surface, except for TKO HeLa cells, which show a slightly lower integrin $\alpha 1$ subunit expression (Fig. R10A). Similarly, all HeLa cells show similar integrin $\alpha 2$ subunit surface expression, except of KKO HeLa cells, which show a higher $\alpha 2\beta 1$ integrin surface expression. The surface expression of the integrin $\beta 1$ subunit is similar amongst all HeLa cell lines, except for KKO HeLa cells, which show a higher surface expression. The surface expression of all tested integrin subunits is similar across all fibroblasts lines, with the exception of KKO and TKO fibroblasts that show slightly higher surface expression levels of the integrin $\alpha 5$ subunit (Fig. R10B).

In summary, the integrin surface expression does not majorly affect the results of SCFS experiments, especially since we are interested in the switch between the canonical and the spatially enhanced adhesion state of cells for which we compare the adhesion within the same cell line. We have included this information in the revised Manuscript (see Supplementary Fig. 11A,B)

Figure R10. Integrin expression level at the cell surface characterized by flow cytometry. Fluorescence intensity of (A) wt HeLa cells, talin KO (TKO) HeLa cells, kindlin KO (KKO) HeLa cells, TKO expressing talin1 head domain (TKO+THD) HeLa cells, and paxillin KO (PXN KO) HeLa cells, and of (B) wt fibroblasts, talin KO (TKO) fibroblasts, kindlin KO (KKO) fibroblasts, TKO expressing talin1 head domain (TKO+THD) fibroblasts, and paxillin KO (PXN KO) fibroblasts. Cells were stained with (A) FITC-integrin $\alpha 1$, FITC-integrin $\alpha 2$ and FITC-integrin $\beta 1$ or (B) biotin-integrin $\alpha 5$ followed by streptavidin-PE, PE-integrin αV and FITC-integrin $\beta 1$ antibody. As negative controls unstained (A) wt HeLa cells and (B) wt fibroblasts were used. $n = 3$ independent experiments for each condition.

Reviewer 1: The TKO results appear rather difficult to be interpreted, since they are sometimes inconsistent between contact times. The authors state that (Line 260) “the adhesion per area was not affected by TKO on smaller patterns”. This seems not fully supported for HeLa cells by the results shown in Fig. 5C showing for some time points significant changes for both, larger and smaller patterns.

Authors: We sincerely thank the reviewer for highlighting that our statements were not fully supported by the data presented. We have carefully revised the Manuscript to town down and adapt the statements accordingly.

Reviewer 1: There are also multiple rather strong statements that seem not to be sufficiently supported by the presented data. To give two examples: 248/249: “... indicates that integrin activation is an essential regulator for sensing the ECM protein area.”

Similarly, it is stated in 341/342: “Hela cells and fibroblasts can measure the size of the substrate area and in response switch between canonical and spatially enhanced adhesion states”

While the results show that cell adhesion is proportionally higher on smaller substrates, and that Mn treatment enhances this effect, it is not clear whether that has to do with active sensing of the ECM contact area.

Authors: We thank the reviewer for pointing out statements that are not fully supported by the data we presented. We have carefully revised the paper to tone down the statements pointed out by the reviewer. We have also toned down statements, which may have not been supported fully by the data.

Reviewer 1: Which effect would lateral diffusion of integrins into the contact zone have, especially at longer contact times. Given that on small patterns the length of the contact zone edge is relatively enhanced, could this also contribute to more recruitment of receptors from nearby regions? It would be great to see this discussed. Also, it might be possible to conduct experiments, e.g. based on confocal/TIRF imaging as mentioned above, or experiments with truncated integrins to explore this effect.

Authors: The reviewer asks which role the lateral diffusion of integrins into the contact zone, especially at longer contact time would have. While conducting SCFS we cannot measure the diffusion of single integrins neither during adhesion initiation nor during the measurement of the adhesion force. However, it has been shown that the diffusion of unbound integrins does not depend on whether they can bind to the substrate²⁴. This indicates that the lateral diffusion of unbound integrins on BSA/FNIII7-10ΔRGD coated substrate areas to the ECM protein patterns may not depend on the ECM protein pattern size. The binding of ligands on ECM protein patterns reduces the free diffusion of integrins. Consequently, the pattern size is expected to show an impact on the overall diffusion of ligand-bound integrins, which may also indirectly impact the diffusion of unbound integrins. However, how such limited diffusion of unbound integrins could affect adhesion initiation during the first 360 s is not clear. Thus, we could only speculate whether and how the diffusive behavior of integrins affects the adhesion initiation on the different ECM protein pattern sizes.

The reviewer also asks whether the length of the contact edge zone (circumference of the ECM protein pattern) relative to the contact area might be relevant for the spatially enhanced adhesion initiation to small ECM protein patterns. It is important that the optical microscopy (epifluorescence or confocal microscopy) combined with our SCFS setup, cannot morphologically/spatially resolve the binding of integrins to the pattern within the first 360 s of adhesion initiation. However, live cell imaging on much longer time scales shows that paxillin-GFP localizes majorly at the circumference of ECM protein patterns (Fig. R4A,B), thus suggesting that the majority of the integrins bind this region of the ECM protein pattern during adhesion initiation. Hence, to evaluate whether the geometry of the ECM protein pattern may influence cells to initiate canonically or spatially enhanced adhesion, we estimated the length of the circumference of each ECM protein pattern size (Fig. R11A). To calculate the circumference of ECM protein patterns,

we used the average area of the pattern and assumed a perfect circular shape (see also Supplementary Fig. 1D, original Manuscript). The results show that the circumference of ECM protein patterns to which cells initiate adhesion canonically are much longer (2- to 6-fold) compared to ECM patterns to which cells initiate spatially enhanced adhesion (Fig. R11A). Hence, in combination with the live cell imaging that localizes paxillin at the circumference of ECM protein patterns, these results indicate that a longer circumference of the ECM protein pattern allows more integrins to bind, which is in line with the cell adhesion force that increases with ECM pattern size (Fig. 3, original Manuscript).

We further analyzed whether there is a correlation between the circumference and area of the ECM protein pattern and the spatially enhanced adhesion (Fig. R11B). The analysis clearly shows that above a circumference to area ratio of 1, cells switch to the spatially enhanced adhesion state, whereas below a circumference to area ratio of 0.7 cells initiate adhesion canonically. Since the fibronectin patterns cover a wider range of circumference to area ratios above 1 (≈ 1.2 to ≈ 2.3) compared to collagen I patterns (≈ 1.7 to ≈ 1.9) might explain the more gradual increase in adhesion force per area of fibroblasts in response to decreasing fibronectin pattern size. However, we can only speculate about the role the circumference to area ratio takes in triggering the cells to switch from the canonical to spatially enhanced adhesion strengthening state. One possible explanation would be that multiple ligand-bound integrins are in closer proximity on smaller ECM patterns that show high circumference to area ratios, and that this proximity accelerates integrin clustering.

In summary, all results suggest that the circumference of the patterns plays a major role in how cells initiate adhesion to ECM protein patterns. We thank the reviewer for this constructive comment and have included the new analysis and information to the revised Manuscript (see Supplementary Fig. 7A,B and in the Results section “*The spatially enhanced adhesion state accelerates adhesion strengthening*” and the Discussion).

Figure R11. The circumference of the ECM protein pattern affects how cells initiate adhesion. (A) Circumference and (B) circumference to area ratio of ECM protein patterns having different areas. Data points present circumferences calculated from average areas of ECM protein patterns (data taken from Fig. 2D, original Manuscript). The boxes indicate to which ECM protein patterns cells initiated adhesion canonically or spatially enhanced (see Discussion section of the revised Manuscript).

Minor:

Reviewer 1: It would be great to see higher resolution images for figure 1E to be able to properly see the adhered cells.

Authors: Thank you. To record higher resolution images for Fig. 1E, we allowed wt HeLa cells or wt fibroblasts to adhere to $\approx 81.2 \mu\text{m}^2$ large collagen I or $\approx 58.1 \mu\text{m}^2$ large fibronectin patterns for 60 min. Afterwards the cells were stained for paxillin and F-actin (Fig. R12). We have added representative images to the revised Fig. 1F and revised the Results section “*Engineering patterned ECM substrates of defined area*” to describe the data.

Figure R12. Confocal images of (A) wt HeLa cells adhering to collagen I pattern or (B) wt fibroblasts adhering to fibronectin pattern. Cells are seeded onto printed Alexa fluor 405-labeled collagen I or fibronectin patterns for 60 min. The printed patterns had areas of $\approx 81.2 \mu\text{m}^2$ (collagen I) and $\approx 58.1 \mu\text{m}^2$ (fibronectin). Afterwards, weakly attached HeLa cells or fibroblasts are removed by gentle washing with PBS. Then, the cells were fixed using 4% (v/v) paraformaldehyde for 15 min at 25°C. 0.2% (v/v) Triton X-100 was used to permeabilize cells for 5 min at 25°C. PBS with 0.1% (v/v) Tween-20 and 3% (w/v) BSA was used to block the sample for 30 min at 25°C. Rabbit anti paxillin followed by donkey anti rabbit Alexa fluor 488 and rhodamine-labeled phalloidin binding to F-actin were used to stain the sample ($n = 10$ independent experiments for each condition). Scale bars, 15 μm .

Reviewer 1: The style the manuscript is written it is somewhat difficult to follow in some parts, especially the part describing the results with talin and paxillin KO cells reads very cumbersome. It would also be helpful, if the motivation for the following sections (e.g. paxillin KO) would be better explained. Also, the abstract could be sharper, e.g. by mentioning a hypothesis or knowledge gap that the paper is addressing.

Authors: We apologize for the unclear presentation of the original Manuscript. We have thoroughly revised it for improved clarity, with particular attention given to the sections concerning talin and paxillin knockout (KO) cells. We have further revised the abstract as suggested to mention the knowledge gap our Manuscript addresses. We hope that the revised Manuscript is now better to follow.

Reviewer 1: In the figures, a plethora of data acquired under different conditions (multiple time points, different reference and multiple patterns) is shown. The visualisation of the plots in Fig. 3-6 is sometimes difficult to read. While it nice to have the partly transparent reference values in the background, it would be good to see the medians in the foreground to compare them to the other medians (some are completely invisible). Also, there are two distinct populations in the normalised data. Did the authors try to perform a cluster analysis and do statistics on the percentage of cells in the two described “states”?

Authors: Thank you, indeed it took us several years to collect sufficient amount of experimental data for our manuscript. We have revised the figures for clarity. We have increased the width of the median bars of the reference data to increase their visibility. However, we would like to abstain from bringing the reference data to the foreground as this becomes confusing to the reader, since it would be even more difficult to distinguish both data sets from each other.

The reviewer also comments on two populations in the normalized data. Please note that we plotted the data on a non-linear axis for better readability of the low adhesion forces per area. This non-linear axis can induce an unintended impression of multiple populations. For clarity we plotted the adhesion forces per area as histograms to check for multiple populations (Fig. R13 and R14). While the histograms for the spatially enhanced adhesion states show a wider distribution of adhesion forces per area compared to the canonical adhesion state, none of the distributions indicates multiple populations.

Figure R13. Adhesion force per area of HeLa cells to printed collagen I patterns having different areas do not show multiple populations. Histograms with a bin width of $0.02 \text{ nN } \mu\text{m}^{-2}$ show the adhesion force per area distribution at the indicated contact time (top) of HeLa cells adhering to collagen I patterns of different areas.

Figure R14. Adhesion forces per area of fibroblasts to printed fibronectin patterns having different areas do not show multiple populations. Histograms with a bin width of $0.1 \text{ nN } \mu\text{m}^{-2}$ show the adhesion force per area distribution at the indicated contact time (top) of fibroblasts adhering to fibronectin patterns of different areas.

Reviewer 1: In Supplementary Figure 5, the mean rupture force is given to describe the distributions. Since most distributions show a truncated Gaussian distribution. Since they can be robustly fit by a Gaussian, it would make sense to also present the modes. In addition, the axis is labelled with counts, so it should probably read relative counts, counts per x or frequency.

Authors: As suggested by reviewer, we calculated the mode of Gaussian distribution and compared the difference between the mean and the mode rupture and tether force (see Table R3). The comparison shows that the mean rupture force and tether force fitted by truncated Gaussian distribution and the mode are very similar. Thus, we decided to keep the mean values to describe distributions.

We thank the reviewer for spotting the wrongly labeled axis of Supplementary Fig. 5. We have revised the y-axis labels in Supplementary Fig. 5 B,C to 'Frequency'.

Mean and mode of rupture and tether force fitted by truncated Gaussian distribution							
Rupture and tether force of HeLa cells							
Patterned area (μm^2)	Unrestricted collagen I	81.2	42.6	33.2	4.3	3.4	Unrestricted BSA
Mean of rupture force (pN)	37.15	37.36	35.35	36.17	35.61	35.21	31.87
Mode of tether force (pN)	37.14	37.35	35.36	36.16	35.62	35.22	31.88
Rupture and tether force of fibroblasts							
Patterned area (μm^2)	Unrestricted fibronectin	58.1	34.1	8.8	5.3	2.4	Unrestricted FNIII7-10 Δ RGD
Mean of rupture force (pN)	44.73	43.40	43.71	43.75	42.84	42.66	36.71
Mode of tether force (pN)	44.73	43.40	43.72	43.75	42.84	42.67	36.71

Table R3. Mean and mode values are similar to describe Gaussian distribution of rupture and tether force. Given are the mean and mode value of Gaussian distribution fitted rupture and tether force of wt HeLa cells and fibroblasts adhering to indicated substrates. Data taken from Supplementary Fig. 5B,C of the original Manuscript.

Reviewer 1: The references are missing. Either this is an issue with PDF version or the authors have submitted an incomplete reference list (references are only till 37 whereas cited number in text goes up to 70).

Authors: We apologize. There must have been a conversion problem. The revised Manuscript now shows the complete list of references.

Reviewer 1: A reference for role of Mn²⁺ in integrin activation should be added.

Authors: The reference had been given in the Methods section “*Single-cell force spectroscopy (SCFS)*” section (Line 479, original Manuscript). Now we have included the reference also in the main text of the revised Manuscript.

Reviewer 1: The authors show a Western blot for the paxillin KO cell lines to show absence of paxillin. Why not show for the talin and kindlin KO cells as well or refer to published data? Also, how do the integrin and talin levels of TKO+THD compare to wildtype cells used as a reference?

Authors: The Western blots showing talin depletion TKO HeLa cells/fibroblasts and kindlin depletion in KKO HeLa cells/fibroblasts are shown in the references that characterize the cell lines in detail (given in the Methods section). We have evaluated the expression of the talin head domain (THD) using western blotting and an antibody that recognizes an epitope in the THD. However, talin antibodies against the THD are rare. The

antibody we have used (Biorad, talin TA205) exclusively recognizes the human talin head domain. To assess the expression levels of the talin head domain, we characterized western blots of wt, TKO and TKO+THD HeLa cells as well as wt, TKO and TKO+THD fibroblasts (Fig. R15). The results show that TKO+THD HeLa cells express lower amounts of THD than wt HeLa cells expressing talin1. These results do not affect our observation that TKO+THD HeLa cells establish similar adhesion force per area to $\approx 81.2 \mu\text{m}^2$ large collagen I patterns and higher adhesion force per area to $\approx 33.2 \mu\text{m}^2$ and $\approx 3.4 \mu\text{m}^2$ large collagen I patterns.

The expression levels of endogenous talin1 in fibroblasts was not accessible, since the human talin head antibody (Biorad, talin TA205) does not recognize mouse talin1. The upper band at ≈ 300 kDa found for wt, TKO, TKO+THD fibroblast in the western blot represents an unspecific band, since TKO fibroblasts do not express talin1. The ≈ 75 kDa band, which is detected in TKO+THD fibroblasts and corresponds to the reintroduced human THD-Ypet. Unfortunately, although antibodies recognizing mouse talin1 are commercially available, to our best knowledge all of them recognize epitopes in the talin rod domain, which makes them unsuitable for our purpose. Hence, we cannot compare the expression levels of talin1 and THD in wt and TKO+THD fibroblasts, respectively. We have included this information in Supplementary Fig. 14A of our revised Manuscript.

Figure R15. Verification of talin expression level in talin1 head domain expressing (TKO+THD) HeLa and fibroblast cell lines. Western blot of cell lysates of wt, talin-depleted (TKO), TKO+THD HeLa cells and wt, talin-depleted (TKO), TKO+THD fibroblasts using anti-talin and anti-glyceraldehyde 3-phosphate dehydrogenase (GAPDH) antibodies. The specific band of full length talin (≈ 270 kDa) was detected in wt HeLa cells and the specific band of human talin head-Ypet (≈ 75 kDa) was detected in TKO+THD HeLa cells and TKO+THD fibroblasts by the anti-talin head antibody. The unspecific band (≈ 300 kDa) was presented in wt, TKO, TKO+THD fibroblasts. GAPDH is shown as a loading control. $n = 3$ independent experiments.

Point-by-Point Response to Reviewers for Manuscript “Mammalian cells measure the extracellular matrix area and respond through switching the adhesion state” NCOMMS-24-20792A-Z

Reviewer #2 (Remarks to the Author)

Reviewer 2: I co-reviewed this manuscript with one of the reviewers who provided the listed reports. This is part of the Nature Communications initiative to facilitate training in peer review and to provide appropriate recognition for Early Career Researchers who co-review manuscripts.

Authors: We thank Reviewer 2 for the thorough review and the encouraging and constructive comments, which guided us to revise and improve our Manuscript.

Point-by-Point Response to Reviewers for Manuscript “Mammalian cells measure the extracellular matrix area and respond through switching the adhesion state” NCOMMS-24-20792A-Z

Reviewer #3 (Remarks to the Author)

Reviewer 3: This is an interesting biophysical approach to looking at how cells respond to small areas of ECM.

Authors: We thank the reviewer for his or her thorough review and the encouraging and constructive comments.

Reviewer 3: As someone who has looked at cells adhering to materials and ECM for decades, I was surprised that fibroblasts don't stick to collagen and HeLa don't stick to fibronectin - I looked at the literature as I was unsure and there are reports of there of both cell types adhering to both ECMs? Why also the differential blocking regime - BSA and RGD depleted FN? I am sure the authors can explain this convincingly.

Authors: Thank you. We did not want to claim that HeLa cells do not adhere to fibronectin. In fact, we have shown before that HeLa cells also adhere to fibronectin^{23,25,26}. However, the fibroblasts we used in this study show very low adhesion force to collagen I, likely due to minimal $\alpha 2\beta 1$ integrin surface expression²⁷. However, due to the time-consuming nature of SCFS experiments (3 to 4 months for a complete data set per cell line and ECM protein), we chose to restrict our characterization to one ECM protein per cell line. We have carefully revised our Manuscript to remove possible confusion in this manner.

We used different blocking proteins since fibroblasts readily interact with BSA. To illustrate this, we seeded fibroblasts onto BSA coated substrates, to which they initiated cell spreading within 60 min, while staying round on FNIII7-10 Δ RGD (Fig. R16A). Further, in SCFS experiments fibroblasts establish drastically higher adhesion forces to BSA compared to FNIII7-10 Δ RGD (Fig. R16B). Hence, we consider BSA as insufficient to passivate the glass surface as fibroblasts somehow adhere to BSA. Further, FNIII7-10 Δ RGD closely mimics fibronectin, without providing specific ligand to which integrins could bind. This lack of binding sites makes FNIII7-10 Δ RGD the ideal for passivation of a surface and to characterize the unspecific adhesion of fibroblasts to fibronectin¹⁶. For HeLa cells we used BSA, as we found previously that HeLa cells interact stronger with FNIII7-10 Δ RGD compared to BSA²⁵. Since FNIII7-10 Δ RGD does not mimic the unspecific adhesion to collagen I, we decided to use BSA as passivation due to the lower unspecific adhesion.

Figure R16. Fibroblasts spread on BSA and fibronectin but stay round on FNIII7-10ΔRGD. (A) Differential interference contrast (DIC) timelapse images of fibroblasts seeded onto FNIII7-10ΔRGD, BSA and fibronectin substrates at indicated time points. $n = 3$ independent experiments for each condition. Scale bars, 15 μm (B) Adhesion force of fibroblasts to BSA at given contact time. Adhesion force of fibroblasts to FNIII7-10ΔRGD is given in the background as reference (semitransparent). Dots represent adhesion forces of individual cells, red bars the median, and $n(\text{cells})$ the number of individual cells tested in at least three independent experiments. P values were calculated by two sided Mann-Whitney tests and compared the adhesion forces of given data with the reference data.

Reviewer 3: I am not a biophysicist, I am a bioengineer/cell biologist - I need to have more information about rupture force. How does it show single integrin events for a whole cell being retracted from the ECM patterns?

Authors: We happily provide additional information how we quantify adhesion relevant metrics from force-distance curves. We can quantify different parameters from force-distance (FD) curves that we record during retracting the cantilever from the support, which mechanically detaches the cell attached to the cantilever from the ECM proteins coating the support^{10,12,13,15}. As shown in a representative FD curve (Fig. R17A, a similar FD curve is shown in Supplementary Fig. 5A, original Manuscript), the most prominent feature is the (maximum) adhesion force of the cell, which is characterized by maximum downward deflection of the cantilever. Upon retracting the cantilever further, the adhesion force is overcome, and the cell starts detaching from the substrate. During the retraction, a complex detachment process of the cell from the substrate can be observed, with multiple rupture events describing the detachment of single or multiple integrins from ECM proteins (Fig. R17B). If the integrin remains anchored to the cell cortex, the rupture force quantifies the maximum force a single integrin can bear before the bond fails. Hence this force difference directly measures the bond properties of individual integrin-ECM protein bonds^{12,13}. If during retraction of the cell from the ECM protein coated support a single or a group of integrins detach from the cell cortex, they separate from the cell surface at the tip of a membrane tether^{12-14,19}. In the FD curve this is shown as force plateau (constant force). The force drop at the end of the plateau characterizes that the membrane tether detaches from the support. However, this force

can also be used to characterize the membrane properties such as membrane tension or fluidity¹⁹.

Fig. R17. Analysis of single-cell force spectroscopy (SCFS) data recorded for a living mammalian cell. (A) Top, The cantilever with the attached cell is approached to a ECM substrate of interest. Upon reaching the preset contact force of 2 nN (in our case), the cantilever height is maintained for the preset contact time. Subsequently, the cantilever is retracted vertically to separate cell and substrate. (A) Bottom, Force–distance (FD) curves recorded during this process, can show different features. From the approach FD curve (red) the apparent cell contact stiffness can be extracted. The retraction FD curve (black) records the force at which the detachment process of the cantilever-attached cell initiates, representing the (maximum) adhesion force of the cell to the substrate. During the mechanical detachment of the cell from the substrate, single receptor (in our case integrin) unbinding events are observed (rupture events). (B) Top, Rupture events, which are recorded when bonds formed between cytoskeleton-linked integrins and the substrate fail, quantify the force a single adhesion receptor can withstand. (B) Bottom, membrane tethers are recorded if a single integrin or multiple integrins at the tip of a membrane tether are mechanically extracted during the separation of cell and substrate. Membrane tethers occur if the integrin linkage to the actomyosin cytoskeleton is either too weak to resist the mechanical load or non-existent. Figure adapted from ref¹³.

Reviewer 3: The authors show that HeLa and fibroblasts increase adhesion, rupture force and adhesion strengthening to smaller ECM islands that larger islands or conformal ECM surfaces. They then look at variants on the cells with talin/kindlin depleted. They showed that adhesion to the smaller ECM islands was not talin of kindlin dependant in contrast to standard understanding of adhesion to large ECM areas. To this point, both cell types agreed well. Finally, the depleted pavilion and added back the talin head and showed the cell types responded differently - effecting specially enhanced adhesion of HeLa cells more so than fibroblasts.

From their data, they conclude that the specially enhanced adhesion state is fundamentally different from canonical adhesion and is not dependant upon mechanotransduction.

Overall, I find these to be interesting but bold claims without backup from imaging data.

I would expect that super-resolution confocal microscopy could help to visualise the specially enhanced mechanism and to make sure that we are not looking at eg adhesion bridging events. Basic things like what do the microfilaments appear like as cells are forced into the specially enhanced state - do stress fibres not form and if mechanotransduction is not involved, does retrograde flow increase?

Authors: The reviewer has correctly summarized our findings on the spatially enhanced adhesion. We thank the reviewer for the encouraging comment and agree with the reviewer that our findings require visual confirmation, which however is only possible at longer contact times. To address how larger adhesion sites assemble, we attached single, rounded paxillin-GFP expressing HeLa cells (Fig. R18A) or fibroblasts (Fig. R18B) to a ConA-coated cantilever and brought the cells into contact with collagen I or fibronectin patterns of different sizes (areas). During contact times of up to ≈ 70 min, we monitored paxillin every 2.5 min by confocal microscopy. In HeLa cells that are in contact with $\approx 81.2 \mu\text{m}^2$ large collagen I patterns for ≈ 7.5 min contact time, paxillin-GFP starts to localize at the periphery of the collagen I pattern. With increasing contact time, the intensity of paxillin-GFP increases both at the periphery and in the center of the pattern. Apparently, more paxillin-GFP localizes at the periphery of the collagen I pattern. On the $\approx 33.2 \mu\text{m}^2$ large collagen I pattern, we observed paxillin-GFP to localize in HeLa cells at the periphery of the pattern, but with slower dynamics. After ≈ 15 min of contact time, HeLa cells adhering to the $\approx 33.2 \mu\text{m}^2$ large collagen I pattern, started to assemble paxillin-GFP only in some peripheral regions. However, the paxillin-GFP covered the entire periphery after ≈ 37.5 min of contact time. No clear increase of fluorescent intensity of paxillin-GFP was observed in the center of the pattern. Interestingly, on the smallest collagen I pattern, we observed paxillin-GFP at some peripheral regions after ≈ 7.5 min, while after ≈ 15 min contact time paxillin-GFP covered the entire periphery of the collagen I pattern.

In fibroblasts, paxillin-GFP localized a bit more diffuse compared to HeLa cells, which is likely due to the low amount of paxillin-GFP on top of the endogenous paxillin expression^{7,8}. However, in fibroblasts paxillin also localized at the periphery of the fibronectin patterns. On the largest fibronectin pattern ($\approx 58.1 \mu\text{m}^2$) paxillin started localizing at the periphery at contact times ≈ 7.5 min. Similar to HeLa cells on collagen I patterns, the localization of paxillin to the intermediate sized fibronectin pattern of $\approx 8.8 \mu\text{m}^2$ was observed from ≈ 22.5 min contact time and hence delayed compared to the largest fibronectin pattern. On the smallest fibronectin pattern of $\approx 2.4 \mu\text{m}^2$, the fibroblasts started recruiting paxillin already at ≈ 7.5 min.

In summary, we observe paxillin recruitment mainly to the periphery of the ECM protein patterns for HeLa cells adhering to collagen I patterns and for fibroblasts adhering to fibronectin patterns. From our timelapse microscopy experiments we conclude that the dynamics of the paxillin recruitment depends on the pattern size. Notably, paxillin recruitment is observed only after much longer contact times than those probed in SCFS experiments (see Fig. 6C,D, original Manuscript). Hence, projecting the results of SCFS to those revealed by confocal microscopy would be rather speculative. Furthermore, the localization of paxillin does not clearly indicate how it influences adhesion, whether it enhances or reduces adhesion forces. Nevertheless, to illustrate how long-term

adhesion, *i.e.*, paxillin recruitment, of HeLa cells and fibroblasts is influenced by the ECM protein pattern size, we have revised our Manuscript to include the data shown in Fig. R6 in Supplementary Fig. 8 and discuss the data in the Results section “*Long term adhesion dynamics depend on the ECM protein pattern area*”.

Figure R18. The formation of adhesion sites by HeLa cells on collagen I patterns and by fibroblasts on fibronectin patterns depends on the pattern size. (A,B) Timelapse confocal microscopy images of paxillin-GFP expressing (A) HeLa cells on collagen I patterns and (B) of fibroblasts on fibronectin patterns. Pattern size (area) and contact time are indicated. $n = 5$ independent experiments for each condition. A single, rounded paxillin-GFP expressing (A) HeLa cell or (B) fibroblast was attached to a ConA-coated AFM cantilever and then brought into contact with collagen I or fibronectin patterns until reaching contact times of ≈ 70 min. Collagen I or fibronectin patterns were labelled using Alexa fluor 555 (gray). Scale bars, $5 \mu\text{m}$.

Further, the reviewer is concerned that the spatially enhanced adhesion could be due to adhesion bridging events, as described by Changede et al²⁸ or Malmström et al²⁹. These publications describe that spatially restricted ECM proteins (or mimetics) that are in proximity ($< 1 \mu\text{m}$ distance between the areas of ECM protein islands) allow cells to form large adhesion sites and induce cell spreading. While these are very interesting observations, we can exclude that cells bridge the ECM patterns to form adhesion sites in our setup because the individual patterns are separated at too far distances from each other. We have carefully designed the ECM protein patterns to have sufficient space between single patterns and to avoid that cells can bridge the patterns. As shown in Supplementary Fig. 1D (original Manuscript), the cantilever attached cells are in contact with a single ECM protein pattern and hence, we can exclude that bridging effects affect our measurements.

The reviewer also asks about the formation of stress fibers or the actin retrograde flow. Both are observable in spread cells but not in rounded cells as characterized in our SCFS experiments. However, we have seeded HeLa cells and fibroblasts on $\approx 81.2 \mu\text{m}^2$ large collagen I or $\approx 58.1 \mu\text{m}^2$ large fibronectin patterns for 1 h, fixed and stained them for paxillin and F-actin (Fig. R19A,B). The results show that even on the largest pattern size, to which the cells initiate adhesion canonically we do not observe the formation of stress fibers. Instead, we observe a homogenous localization of paxillin and actin at the circumference of the ECM protein pattern. This is likely due to the restriction of ECM protein, which does not allow the cell to polarize. We have added representative images to the revised Fig. 1F.

Figure R19. Confocal images of (A) wt HeLa cells adhering to collagen I pattern or (B) wt fibroblasts adhering to fibronectin pattern. Cells are seeded onto printed Alexa fluor 405-labeled collagen I or fibronectin patterns for 60 min. The printed patterns had areas of $\approx 81.2 \mu\text{m}^2$ (collagen I) and $\approx 58.1 \mu\text{m}^2$ (fibronectin). Afterwards, weakly attached HeLa cells or fibroblasts are removed by gentle washing with PBS. Then, the cells were fixed using 4% (v/v) paraformaldehyde for 15 min at 25°C. 0.2% (v/v) Triton X-100 was used to permeabilize cells for 5 min at 25°C. PBS with 0.1% (v/v) Tween-20 and 3% (w/v) BSA was used to block the sample for 30 min at 25°C. Rabbit anti paxillin followed by donkey anti rabbit Alexa fluor 488 and rhodamine-labeled phalloidin binding to F-actin were used to stain the sample. $n = 10$ independent experiments for each condition. Scale bars, 15 μm .

To investigate whether mechanotransduction is involved in the canonical and spatially enhanced adhesion initiation, we performed SCFS experiments of HeLa cells and fibroblasts in the presence of 20 μM blebbistatin to perturb myosin II contractility. We quantified the cell adhesion force to three different sizes of collagen I patterns ($\approx 81.2 \mu\text{m}^2$, $\approx 33.2 \mu\text{m}^2$, and $\approx 3.4 \mu\text{m}^2$, Fig. R20A,B) or fibronectin patterns ($\approx 58.1 \mu\text{m}^2$, $\approx 8.8 \mu\text{m}^2$, and $\approx 2.4 \mu\text{m}^2$, Fig. R20C,D). Inhibition of myosin II lowered the adhesion force of HeLa cells to collagen I patterns having an area of $\approx 81.2 \mu\text{m}^2$ at contact times ≥ 120 s (Fig. R20A). However, the adhesion force of HeLa cells to the smaller patterns remained unaffected by the inhibition of myosin II. Hence, the adhesion force per area of HeLa cells to $\approx 81.2 \mu\text{m}^2$ large collagen I patterns is slightly decreased at contact times ≥ 120 s (Fig. R20B). These results show that the canonical adhesion initiation to collagen I depends on myosin II-mediated actin contractility at contact times ≥ 120 s, while the spatially enhanced adhesion of HeLa cells to collagen I is independent of actomyosin contractility within the first 360 s of contact.

Inhibition of myosin II reduced the adhesion force of fibroblasts to $\approx 58.1 \mu\text{m}^2$ large fibronectin patterns for contact times ≥ 20 s and, surprisingly, also for the smallest ($\approx 2.4 \mu\text{m}^2$) fibronectin patterns for all contact times (Fig. R20C). In contrast, the adhesion force of fibroblasts to the $\approx 8.8 \mu\text{m}^2$ large fibronectin patterns remained unchanged. Hence,

also the adhesion force per area to the largest and smallest fibronectin pattern reduced, while it remained similar to the intermediate size fibronectin pattern (Fig. R20D). These results are in good agreement with those obtained using TKO+THD fibroblasts, which showed a similar behavior (see Fig. 6B, original Manuscript).

Taken together, the additional experimental results strengthen the conclusions made in the manuscript and add that actomyosin contractility and hence mechanotransduction is cell line and/or integrin type specific and involves in regulating the spatially enhanced cell adhesion. We have included and discussed the additional experimental data in the revised Manuscript (see revised Fig. 6A,B, Supplementary Fig. 10A,B, Supplementary Fig. 12, and Supplementary Fig. 13A,B, Methods section “Single-cell force spectroscopy (SCFS)”, and Results section “Cell-specific integrin-actin engagement and integrin-mediated ECM sensing”).

Figure R20. The spatially enhanced adhesion state is independent of myosin II contractility. (A-D) Adhesion force and adhesion force per area of 20 μM blebbistatin-treated (A,B) wt HeLa cells and (C,D) wt fibroblasts to different areas of collagen I or fibronectin patterns at given contact times. Dots represent adhesion forces of individual cells, red bars the median, and *n*(cells) the number of individual cells tested in at least three independent experiments. (A,C) Adhesion force and (B,D) adhesion force per area of untreated wt HeLa cells or wt fibroblasts are given as reference (semitransparent). *P* values were calculated by two sided Mann-Whitney tests and compare the given adhesion forces with the reference data.

Reviewer 3: It would also be interesting to try fibroblasts adhering to fibronectin vs vitronectin, where adhesion bridging is more regularly seen and where the cells can form larger adhesions in response to restricted ECM patterns (as shown in work by Duncan Sutherland and others).

Authors: The reviewer suggests probing the adhesion to restricted vitronectin patterns to observe, whether adhesion bridging plays a role on the spatially enhanced adhesion. However, as described above, bridging does not play a role in our experimental setup because the printed ECM patterns are at too far distance. Further, while adding another ECM protein would be surely interesting, it requires a significant amount of experimental time as described above, which would be beyond scope of the revisions. However, inspired by the reviewers suggestion, we are thinking of designing a different experimental setup in which we engineer the printed patterns closer so that adhesion bridging can occur and which we can use compare the different roles fibronectin and vitronectin take in triggering adhesion bridging.

Reviewer 3: I hope that the manuscript has attracted insightful reviews from more biophysical reviewers than myself - but, for me, I would like to see the paper including more classical cell biology for confirmation.

Authors: We again thank the reviewer for the constructive comments, which guided us to revise and further improve our Manuscript.

References

1. Sun, Z., Costell, M. & Fässler, R. Integrin activation by talin, kindlin and mechanical forces. *Nat. Cell Biol.* **21**, 25–31 (2019).
2. Calderwood, D. A. *et al.* The Talin head domain binds to integrin beta subunit cytoplasmic tails and regulates integrin activation. *J. Biol. Chem.* **274**, 28071–28074 (1999).
3. Klapholz, B. & Brown, N. H. Talin – the master of integrin adhesions. *J. Cell Sci.* **130**, 2435–2446 (2017).
4. Moser, M., Legate, K. R., Zent, R. & Fässler, R. The tail of integrins, talin, and kindlins. *Science* **324**, 895–899 (2009).
5. Yao, M. *et al.* The mechanical response of talin. *Nat. Commun.* **7**, 11966 (2016).
6. Böttcher, R. T., Strohmeyer, N., Aretz, J. & Fässler, R. New insights into the phosphorylation of the threonine motif of the $\beta 1$ integrin cytoplasmic domain. *Life Sci Alliance* **5**, e202101301 (2022).
7. Ahn, S., Sharma, U., Kasuba, K. C., Strohmeyer, N. & Müller, D. J. Engineered Biomimetic Fibrillar Fibronectin Matrices Regulate Cell Adhesion Initiation, Migration, and Proliferation via $\alpha 5\beta 1$ Integrin and Syndecan-4 Crosstalk. *Adv. Sci.* **10**, e2300812 (2023).
8. Spoerri, P. M., Strohmeyer, N., Sun, Z., Fässler, R. & Müller, D. J. Protease-activated receptor signalling initiates $\alpha 5\beta 1$ -integrin-mediated adhesion in non-haematopoietic cells. *Nat. Mater.* **19**, 218–226 (2020).
9. Taubenberger, A., Cisneros, D. A., Puech, P.-H., Müller, D. J. & Franz, C. M. Revealing early steps of $\alpha 2\beta 1$ integrin-mediated adhesion to collagen type I by using single-cell force spectroscopy. *Mol. Biol. Cell* **18**, 1634–1644 (2007).
10. Friedrichs, J. *et al.* A practical guide to quantify cell adhesion using single-cell force spectroscopy. *Methods* **60**, 169–178 (2013).
11. Taubenberger, A. V., Hutmacher, D. W. & Müller, D. J. Single-Cell Force Spectroscopy, an Emerging Tool to Quantify Cell Adhesion to Biomaterials. *Tissue Engineering Part B: Reviews* **20**, 40–55 (2014).
12. Viljoen, A. *et al.* Force spectroscopy of single cells using atomic force microscopy. *Nat Rev Methods Primers* **1**, 1–24 (2021).
13. Müller, D. J., Helenius, J., Alsteens, D. & Dufrêne, Y. F. Force probing surfaces of living cells to molecular resolution. *Nat. Chem. Biol.* **5**, 383–390 (2009).

14. Riet, J. T. *et al.* Dynamic coupling of ALCAM to the actin cortex strengthens cell adhesion to CD6. *J. Cell Sci.* **127**, 1595–1606 (2014).
15. Helenius, J., Heisenberg, C.-P., Gaub, H. E. & Müller, D. J. Single-cell force spectroscopy. *J. Cell Sci.* **121**, 1785–1791 (2008).
16. Benito-Jardón, M. *et al.* α V-Class integrin binding to fibronectin is solely mediated by RGD and unaffected by an RGE mutation. *J. Cell Biol.* **219**, 507 (2020).
17. Strohmeyer, N., Bharadwaj, M., Costell, M., Fässler, R. & Müller, D. J. Fibronectin-bound α 5 β 1 integrins sense load and signal to reinforce adhesion in less than a second. *Nat. Mater.* **16**, 1262–1270 (2017).
18. Bharadwaj, M. *et al.* α V-class integrins exert dual roles on α 5 β 1 integrins to strengthen adhesion to fibronectin. *Nat. Commun.* **8**, 14348 (2017).
19. Krieg, M., Helenius, J., Heisenberg, C.-P. & Müller, D. J. A Bond for a Lifetime: Employing Membrane Nanotubes from Living Cells to Determine Receptor-Ligand Kinetics. *Angew. Chem. Int. Ed.* **47**, 9775–9777 (2008).
20. Hochmuth, R. M. & Marcus, W. D. Membrane Tethers Formed from Blood Cells with Available Area and Determination of Their Adhesion Energy. *Biophys. J.* **82**, 2964–2969 (2002).
21. Friedrichs, J., Helenius, J. & Müller, D. J. Quantifying cellular adhesion to extracellular matrix components by single-cell force spectroscopy. *Nat. Protoc.* **5**, 1353–1361 (2010).
22. Theodosiou, M. *et al.* Kindlin-2 cooperates with talin to activate integrins and induces cell spreading by directly binding paxillin. *eLife* **5**, e10130 (2016).
23. Huber, M., Casares-Arias, J., Fässler, R., Müller, D. J. & Strohmeyer, N. In mitosis integrins reduce adhesion to extracellular matrix and strengthen adhesion to adjacent cells. *Nat Commun* **14**, 2143 (2023).
24. Rossier, O. *et al.* Integrins β 1 and β 3 exhibit distinct dynamic nanoscale organizations inside focal adhesions. *Nat. Cell Biol.* **14**, 1057–1067 (2012).
25. Schubert, R. *et al.* Assay for characterizing the recovery of vertebrate cells for adhesion measurements by single-cell force spectroscopy. *FEBS Lett.* **588**, 3639–3648 (2014).
26. Yu, M., Strohmeyer, N., Wang, J., Müller, D. J. & Helenius, J. Increasing throughput of AFM-based single cell adhesion measurements through multisubstrate surfaces. *Beilstein J. Nanotechnol.* **6**, 157–166 (2015).

27. Schiller, H. B. *et al.* β 1- and α v-class integrins cooperate to regulate myosin II during rigidity sensing of fibronectin-based microenvironments. *Nat. Cell Biol.* **15**, 625–636 (2013).
28. Changede, R., Cai, H., Wind, S. J. & Sheetz, M. P. Integrin nanoclusters can bridge thin matrix fibres to form cell–matrix adhesions. *Nat Mater* **18**, 1366–1375 (2019).
29. Malmström, J. *et al.* Focal Complex Maturation and Bridging on 200 nm Vitronectin but Not Fibronectin Patches Reveal Different Mechanisms of Focal Adhesion Formation. *Nano Lett.* **11**, 2264–2271 (2011).

Reviewer #1 (Remarks to the Author)

Reviewer 1: In the rebuttal letter and revised manuscript, the authors have thoroughly addressed all comments point-by-point.

Authors: Thank you for your positive recognition of our revision. Below, we provide detailed responses to the remaining points raised by the reviewer and outline the corresponding revisions made to the Manuscript.

Reviewer 1: With regards to my previous questions on the analysis: it was good to see a direct comparison of the original and proposed alternative analysis method, e.g. for single unbinding events, adhesion work etc. Since the shown differences do apparently not significantly influence the key findings and the limitations are also discussed in the manuscript now, there are not any further queries related to the analysis from my side.

Authors: Thank you.

Reviewer 1: Also, additional experiments were undertaken that strengthen the paper in my opinion. Importantly, the results of the myosin blocking experiments are interesting and support the original findings. I am confused, however, with the interpretation of the results in Fig. 6B (Fig. R1C). The authors state in the rebuttal letter “Hence, also the adhesion force per area to the largest and smallest fibronectin pattern reduced, while it remained similar to the intermediate sized fibronectin pattern (Fig. R1D)”, but (despite the shown p-values) the medians of blebbistatin-treated and controls for 2.4 μ m² in R1D look rather similar to untreated controls? Please check the datasets again and also revise the figure legend of Fig. 6 to make it clear what the respective reference measurements in A and B are (I assume the legend should read untreated WT cells as in Fig. R1?).

Authors: Thank you for pointing out this mistake. We apologize for having created confusion. We indeed did not compare the correct data sets in our statistical analysis. Hence, we have reanalyzed the data and now provide the correct *P* values in the revised Fig. 6B and Supplementary Fig. 13B (below shown as Fig. R1). However, the overall statement remains correct: The blebbistatin-treatment reduces the adhesion force per area of wt fibroblasts to the smallest fibronectin patterns, with the exception of 5 s and 240 s contact time. We understand, that the adhesion force per area look very similar. However, this is likely a scaling issue of the upper and lower part of the force axis (y-axis). We have revised the Results section “*Cell-specific integrin-actin engagement and integrin-mediated ECM sensing*” accordingly. Additionally, we have revised the legend of Fig. 6 to improve clarity. The revised sentence of the figure legend now reads “*Adhesion forces per area of untreated wt HeLa cells or wt fibroblasts in the respective condition are given as reference (semitransparent)*”.

Fig. R1, included as revised Supplementary Fig. 13B and Fig. 6B of the manuscript. (A) Adhesion force and (B) adhesion force per area of 20 μM blebbistatin-treated wt fibroblasts to different areas of fibronectin patterns at given contact times. (A) Adhesion force and (B) adhesion force per area of untreated wt fibroblasts are given as reference (semitransparent). Dots represent adhesion forces of individual cells, red bars the median, and $n(\text{cells})$ the number of individual cells tested in at least three independent experiments. P values were calculated by two sided Mann-Whitney tests and compare the given adhesion forces with the reference data.

Reviewer 1: Moreover, the imaging data provide interesting insights into the kinetics of adhesion formation in dependence of the adhesion pattern size. This, together with the more detailed analysis of the single unbinding events, improves interpretation of the findings.

Authors: Thank you again for your encouraging comment.

Reviewer 1: Line 484 is confusing to me. If there are no clear signs of cluster formation within the analysed attachment period of the SCFS experiments, why do you conclude that clustering is one of the driving mechanisms? Although it might be true, this should be better discussed.

Authors: The reviewer is correct - due to the diffraction limit of the optical microscope used we cannot optically resolve integrin cluster formation within the contact time used in the SCFS experiments. However, it is possible that integrin clustering is accelerated in the spatially enhanced adhesion state. We thus originally concluded from the different effects, which actomyosin contractility inhibition and paxillin depletion show in HeLa cells and fibroblasts (Fig. 6A,B,E,F, original Manuscript) that “*The results, thus, indicate that actomyosin contractility, integrin clustering, and adhesome formation play cell and/or integrin type specific roles in sensing the ECM protein area and triggering the cellular response to switch the adhesive state*” (original sentence of the Manuscript; line 484). Both, actomyosin contractility and paxillin are major regulators of integrin clustering. To avoid confusion of the reviewer the wording “integrin clustering” has been removed from above cited sentence. The revised sentence now reads “*The results, thus, indicate that actomyosin contractility and adhesome formation play cell and/or integrin type specific roles in sensing the ECM protein area and triggering the cellular response to switch the adhesive state*”.

Reviewer 1: Generally, it remains still unclear to me whether one can confidently talk about an active sensing mechanism (e.g. "cells sense and respond", line 423), in particular when taking the relevance of the adhesion patterns circumference into account, which is nicely shown now. Nevertheless, the observation of a spatially enhanced adhesion is of interest, and might be even relevant for the discussed scenario of a native ECM environment.

Authors: Thank you for your encouraging comment on the general importance of the spatially enhanced adhesion in the scenario of a native ECM. We agree that referring to "cells sense and respond" could be misleading, especially given how cell adhesion to confined ECM appears to switch based on pattern circumference rather than classic focal adhesion formation. We have revised the Manuscript accordingly and the revised sentence reads "*Importantly, the results also highlight that HeLa cells and fibroblasts respond to decreasing ECM protein area by initiating adhesion differently*".

Reviewer 1: While it is understandable that a comparison of talin levels in wt and TKO-THD fibroblasts is technically difficult, it would still be worth to present data confirming absence of talin in the talin KO fibroblasts using another antibody.

Authors: As suggested by reviewer, we used another set of antibodies, anti talin 1,2 (8D4 from Sigma, Fig. R2A), anti talin1 (97H6 from Abcam, Fig. R2B), and anti talin2 (68E7 from Abcam, Fig. R2C), to detect the expression level of talin 1 and talin 2 in wt fibroblasts and talin-depleted (TKO) fibroblasts. The specific band of talin \approx 270 kDa was detected in wt fibroblasts but not detectable in TKO fibroblasts, thus verifying the talin knock out in TKO fibroblasts. Our western blot confirms the data published earlier by Theodosiou *et al.* on the TKO fibroblasts¹. We have included the western blot and the corresponding reference¹ into our revised Manuscript (see Supplementary Fig. 14).

Figure R2, included in new Supplementary Fig. 14 of the revised Manuscript. Verification of talin expression level in talin-depleted (TKO) fibroblast cell lines. Western blot of cell lysates of wt fibroblasts, talin-depleted (TKO) fibroblasts, and TKO expressing talin1 head domain (TKO + THD) fibroblasts using anti-talin1,2 (1:100, monoclonal anti-talin antibody produced in mouse, T3287, Sigma), anti-talin1 (1:500, anti-talin 1 antibody [97H6], ab108480, Abcam), and anti-talin2 antibodies (1:500, anti-talin 2 antibody [68E7], ab105458, Abcam). All samples were also labelled with anti-glyceraldehyde 3-phosphate dehydrogenase (GAPDH) antibodies (1:2500, anti-GAPDH, ab9485, Abcam), as a loading control. The specific band of full length talin (≈ 270 kDa) is only detected in wt fibroblasts. A weak band at ≈ 230 kDa is detected for TKO fibroblasts¹ and TKO + THD fibroblasts (kindly provided by C. Grashoff, University of Münster, Germany), which are derived from different sources (Methods). Because this band does not differentiate between both cell lines, it is considered unspecific. $n = 3$ independent experiments.

Reviewer 1: Typo line 290 (dependent).

Authors: Thank you, we have corrected the typo.

Point-by-Point Response to Reviewers for Manuscript “Mammalian cells measure the extracellular matrix area and respond through switching the adhesion state”
NCOMMS-24-20792B

Reviewer #2 (Remarks to the Author)

Reviewer 2: I co-reviewed this manuscript with one of the reviewers who provided the listed reports. This is part of the Nature Communications initiative to facilitate training in peer review and to provide appropriate recognition for Early Career Researchers who co-review manuscripts.

Authors: We thank the reviewer for the thorough review and the encouraging and constructive comments, which helped us to further improve our manuscript.

Point-by-Point Response to Reviewers for Manuscript “Mammalian cells measure the extracellular matrix area and respond through switching the adhesion state”
NCOMMS-24-20792B

Reviewer #3 (Remarks to the Author)

Reviewer 3: Reviewer 1 and 2 combined have produced a range of highly appropriate biophysical comments to which the authors have responded to at length. I found their review very useful to myself.

I was not strongly supportive on first review, but I feel the authors have answered my questions as best they can and explained limitations of what they can achieve well.

If reviewers' 1 and 2 are happy with the response to their questions, I am satisfied.

Authors: We thank the reviewer for the thorough review and the encouraging and constructive comments, which helped us to further improve our manuscript.

References

1. Theodosiou, M. *et al.* Kindlin-2 cooperates with talin to activate integrins and induces cell spreading by directly binding paxillin. *eLife* **5**, e10130 (2016).